# Two-dimensional single-crystalline mesoporous high-entropy oxide nanoplates for efficient electrochemical biomass upgrading

Yanzhi Wang[1], Hangjuan He[1], Hao Lv[1,2], Fengrui Jia[1] & Ben Liu [1] ✉

Mesoporous single crystals have received more attention than ever in catalysis-related applications due to their unique structural functions. Despite great efforts, their progress in engineering crystallinity and composition has been remarkably slower than expected. In this manuscript, a template-free strategy is developed to prepare two-dimensional high-entropy oxide (HEO) nanoplates with single-crystallinity and penetrated mesoporosity, which further ensures precise control over high-entropy compositions and crystalline phases. Single-crystalline mesoporous HEOs (SC-MHEOs) disclose high electrocatalytic performance in 5-hydroxymethylfurfural oxidation reaction (HMFOR) for efficient biomass upgrading, with remarkable HMF conversion of 99.3% and superior 2,5-furandicarboxylic acid (FDCA) selectivity of 97.7%. Moreover, with nitrate reduction as coupling cathode reaction, SC-MHEO realizes concurrent electrosynthesis of value-added FDCA and ammonia in the two-electrode cell. Our study provides a powerful paradigm for producing a library of novel mesoporous single crystals for important catalysis-related applications, especially in the two-electrode cell.

Catalytic conversion of renewable non-fossil biomass-derived chemicals is an efficient upgrading route to obtain high value-added products[1,2]. As an important dehydration product from cellulosic biomass, for example, 5-hydroxymethylfurfural (HMF) has exhibited great potential to produce value-added feedstocks[3,4]. Especially, 2,5-furandicarboxylic acid (FDCA), as one of the twelve priority compounds in green chemical industries (bio-plastics and fine chemicals), can be directly conversed by selective electrooxidation of HMF[5]. Traditionally, selective HMF-to-FDCA conversion is performed on a liquid-phase reaction, which generally requires harsh conditions, including the utilization of noble metal catalysts (Pt, Pd, Au, etc.) and toxic oxidants ($KMnO_4$, $K_2Cr_2O_7$, etc.) as well as high temperature (100–200 °C) and pressure (5–10 atm)[6,7]. This results remarkably in higher cost and energy consumption in the chemical industry. In sharp comparison, electrocatalytic HMF-to-FDCA reaction is an environmentally benign and economic alternative route, the process of which is driven by electrons at the anode with $H_2O$ as the oxidant[8-10]. Despite great efforts over recent years, their performance in both activity and stability is still unsatisfactory than expected, specifically considering its coupling cathode hydrogen evolution reaction (HER) due to the lack of high-performance noble metal-free electrocatalysts.

Single-crystalline mesoporous transition metal oxides contribute a new class of nanostructured materials and have behaved their wide utilizations in catalysis and electrocatalysis[11-14]. Long-range structural coherences of oxide single crystals remarkably accelerate the transports of electrons and reactants, while their penetrated mesopores expose abundant catalytically active sites[15-20]. These structural synergies thus meet theoretically the demand in catalytic materials with

[1]Key Laboratory of Green Chemistry and Technology of Ministry of Education, College of Chemistry, Sichuan University, 610064 Chengdu, China. [2]School of Chemistry and Chemical Engineering, Shanghai Jiao Tong University, 200240 Shanghai, China. ✉e-mail: ben.liu@scu.edu.cn

defined functionality and enhanced performance[21–25]. However, the synthesis of single-crystalline mesoporous transition metal oxides is highly challenging, mostly because the crystallization and growth of grain oxides result thermodynamically in remarkable volume shrinkage and structure collapse[26–30]. Therefore, most of the mesoporous transition metal oxides reported in the literature are low-crystalline and/or polycrystalline, which potentially impedes their theoretical investigations and practical applications to some extent[31–34].

On the other hand, high-entropy oxides (HEOs), which are composed of at least five transition metals arranged in a single-phase solid solution with homogeneous yet random distributions, have received more attention than ever compared to their monometallic counterparts[35–41]. First, multimetallic elements of HEOs ensure more compositional diversities and structural complexities, thus producing unprecedented geometric and electronic properties[8,42–46]. Meanwhile, the difference in metal atomic sizes causes severe lattice distortions, which would optimize the energy barriers for various reactant/intermediate/product molecules[47–51]. Therefore, HEOs have generally exhibited much better activity and desired selectivity for their wide utilizations from catalysis and electrocatalysis to energy storage and conversion[52–54]. Second, the entropy-stabilized single-phase state of HEOs is more thermodynamically favorable, which could obey the physical sintering process and remarkably enhance their (electro)catalytic stability[55–58]. Inspired by the above discussion, we rationally anticipate a novel high-performance (electro)catalyst material that combines multiple advantages of HEO composition and single-crystallinity as well as penetrated mesoporosity for selective HMF-to-FDCA electrocatalysis. However, the formation of HEOs generally requires high temperature, leading thermodynamically to a huge loss of surface area and producing particle-like materials (P-HEO) with uncontrolled nanostructures and poor crystallinity (Fig. 1a and Supplementary Fig. S1). To the best of our knowledge, low-dimensional single-crystalline mesoporous HEOs (SC-MHEOs) with well-defined nanostructures and controlled metal compositions have never been achieved thus far, leaving a big room to explore their potential application in selective HMF-to-FDCA electrocatalysis.

In this work, we report a template-free synthetic strategy for preparing uniform and high-purity two-dimensional SC-MHEO nanoplates. The synthesis relies on the direct conversion of single-crystalline high-entropy basic carbonate salts (SC-HEBCSs) under high temperature, in which the release of $H_2O$ and $CO_2$ during the crystallization produces abundant and penetrated mesopores and retains single-crystalline oxide frameworks in the absence of any templates. Four $Co_3O_4$-like SC-MHEOs, including quinary $(CoNiMnCuZn)_3O_4$ and $(CoNiMnCuFe)_3O_4$, senary $(CoNiMnCuZnBi)_3O_4$, and septenary $(CoNiMnCuZnFeBi)_3O_4$, and one CoO-like SC-MHEO $((CoNiMnCuZn)O)$ are prepared with well-defined two-dimensional nanoplate nanostructure. When being performed as an electrocatalyst for biomass upgrading, two-dimensional SC-MHEO nanoplates disclose superior performance in selective HMF oxidation reaction (HMFOR) for FDCA electrosynthesis under mild conditions. Compared with other counterparts, the best SC-MHEO-$(CoNiMnCuZn)_3O_4$ holds a remarkable HMF conversion of 99.3%, a superior FDCA selectivity of 97.7%, and a high consecutive cycle stability. Moreover, the two-electrode coupling system, by replacing HER with nitrate reduction reaction ($NO_3^-RR$) at the cathode, concurrently produces value-added FDCA and ammonia ($NH_3$) under a low energy consumption.

## Results

Different from the traditional high-temperature treatment of metal precursors, our synthesis of two-dimensional SC-MHEOs requires two separate steps, including the preparation of two-dimensional SC-HEBCSs and the high-temperature conversion of BCSs into oxide (BCS-oxide) in the absence of any templates (Fig. 1b). In a typical synthesis of SC-HEBCSs, metal nitrates (for example, $Co^{2+}$, $Ni^{2+}$, $Mn^{2+}$, $Cu^{2+}$, and $Zn^{2+}$) were first mixed with urea in water/ethanol containing oleic acid as a stabilizer to form a homogeneous solution. After being hydrothermally treated under 160 °C, SC-HEBCS-$(CoNiMnCuZn)_2(OH)_2CO_3$ nanoplates with two-dimensional morphology and high-entropy composition were prepared accordingly. Powder X-ray diffraction (XRD) pattern shows that SC-HEBCS-$(CoNiMnCuZn)_2(OH)_2CO_3$ discloses a monoclinic crystalline structure with a P21/a space group (PDF 01-079-7085), which is completely same to the crystalline structure of monometallic $Co_2(OH)_2CO_3$ (Supplementary Fig. S2a)[59]. Scanning electron microscopy (SEM) and transmission electron microscopy (TEM) images further show that both SC-HEBCS-$(CoNiMnCuZn)_2(OH)_2CO_3$ and $Co_2(OH)_2CO_3$ are highly uniform and homogeneous with a two-dimensional plate-like nanostructure (Supplementary Fig. S2b−e). The average length, center width, and thickness are determined as 3.1 μm, 480 nm, and 90 nm, respectively (Supplementary Fig. S3). Selected area electron diffraction (SAED)

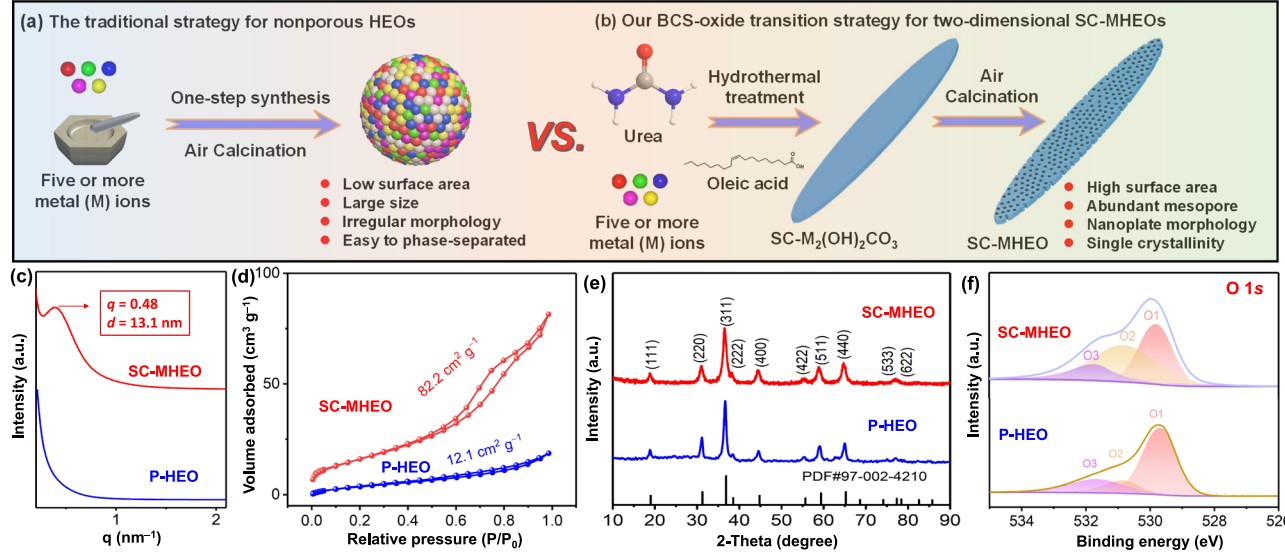

**Fig. 1 | Synthesis strategy. a** Traditional synthesis method and **b** BCS-oxide transition strategy for HEOs. **c** SAXS patterns, **d** $N_2$ sorption isotherms, **e** wide-angel XRD patterns, and **f** high-resolution XPS 1s spectra of SC-MHEO and P-HEO.

pattern of a sample discloses a single set of signals, indicating the product is single-crystalline with (010) exposed facet (Supplementary Fig. S4), which is further indicated by high-resolution TEM image (Supplementary Fig. S5). High-angle annular dark-field scanning TEM (HAADF-STEM) energy dispersive spectroscopy (EDS) element mapping images further demonstrate highly distributed Co, Ni, Mn, Cu, and Zn elements within two-dimensional nanoplate (Supplementary Fig. S6). The Co/Ni/Mn/Cu/Zn atomic ratio of $(CoNiMnCuZn)_2(OH)_2CO_3$ is 35.0/27.0/15.9/13.6/8.5, which is identical to the ratio obtained from inductively coupled plasma-mass spectrometry. These results confirm the successful synthesis of two-dimensional SC-HEBCSs that ensures a solid platform for the solid-phase formation of SC-MHEOs under controlled conditions.

Two-dimensional SC-MHEO nanoplates were then prepared by directly treating SC-HEBCS-$(CoNiMnCuZn)_2(OH)_2CO_3$ at a high temperature (300 °C) under air atmosphere in the absence of any templates. Small-angle X-ray scattering (SAXS) patterns reveal that SC-MHEO shows a characteristic peak at $q$ value of 0.48, which is almost same to monometallic single-crystalline mesoporous $Co_3O_4$ (SC-M-$Co_3O_4$) (Supplementary Fig. S7), corresponding to an average mesoporous periodicity of 13.1 nm (Fig. 1c). In contrast, P-HEO synthesized by direct calcination of metal precursors is structurally nonporous, which has a Co/Ni/Mn/Cu/Zn atomic ratio of 33.5/26.8/16.6/15.9/7.2 (Supplementary Table S2). Meanwhile, nitrogen ($N_2$) sorption isotherms of SC-MHEO show a pore size of 3–8 nm, further indicating mesoporous structure (Supplementary Fig. S8). The Brunauer–Emmett–Teller (BET) surface area of SC-MHEO is 82.2 m$^2$ g$^{-1}$ due to the presence of abundant mesopores, which reaches 6.8 times higher than that of nonporous P-HEO (12.1 m$^2$ g$^{-1}$) (Fig. 1d). Powder XRD patterns exhibit a single type of diffraction signals for SC-MHEO and P-HEO (Fig. 1e), which are same to the signals for SC-M-$Co_3O_4$ (Supplementary Fig. S9), confirming both of them have a single-phase spinel crystalline structure (PDF: 97-002-4210). In comparison to monometallic SC-M-$Co_3O_4$, however, all the peaks of SC-MHEO slightly shift towards the lower degrees, confirming that the Co sites in monometallic $Co_3O_4$ are randomly occupied by other four metals (Ni, Mn, Cu, and Zn) to form a HEO in the single-phase spinel structure. HEOs with homogeneous composition and $Fd\bar{3}m$ crystalline structure are further confirmed by corresponding Raman spectra (Supplementary Fig. S10)[60,61]. Besides, X-ray photoelectron spectroscopy (XPS) was used to study the electronic state of anionic oxygen in HEOs. High-resolution O 1$s$ spectra are deconvoluted into three peaks at 529.7 (O1, lattice oxygen), 530.9 (O2, defected or uncoordinated oxygen), and 531.9 eV (O3, adsorbed oxygen) (Fig. 1f)[8,62]. Compared with P-HEO, obviously, more O2 species is obtained for SC-MHEO, indicating rich mesoporous channels produce more defected and uncoordinated oxygen sites for electrocatalysis.

The morphology and nanostructure of SC-MHEO are further characterized by various advanced electron microscopies. SEM image shows the high quality and homogeneity of SC-MHEO with a two-dimensional plate-like nanostructure (Fig. 2a), which is almost the same as its parent high-entropy $(CoNiMnCuZn)_2(OH)_2CO_3$. In comparison to parent SC-HEBCS, SC-MHEO becomes slightly smaller with average length, center width, and thickness of 2.9 μm, 460 nm, and 75 nm, indicating a minor volume shrinkage during the BCS-oxide transition process (Fig. 2b and Supplementary Fig. S11). Meanwhile, the atomic force microscope (AFM) image of SC-MHEO showed a typical two-dimensional plate-like morphology with an average thickness of approximately 78 nm (Supplementary Fig. S12). HAADF-STEM image of a single SC-MHEO further shows a two-dimensional nanostructure (Fig. 2c), which is the same as monometallic SC-M-$Co_3O_4$. There are abundant and penetrated mesopores throughout the nanoplate with mesopores size of 3–10 nm, confirming they are structurally mesoporous (Fig. 2d, e). HAADF-STEM EDS mapping images clearly show uniform distributions (no element segregation) of five metal and O

elements throughout the nanoplate (Fig. 2f). The Co/Ni/Mn/Cu/Zn atomic ratio is 34.8/28.1/16.0/13.2/7.9 (Supplementary Table S1), which is consistent with its parent $(CoNiMnCuZn)_2(OH)_2CO_3$, indicating they are compositionally high-entropy.

High-resolution TEM image and corresponding Fourier transform (FT) pattern show a clear and uniform interlayer spacing distance of 0.285 nm, which is slightly larger than monometallic SC-M-$Co_3O_4$, corresponding to the (220) crystalline phase (Fig. 3a–c). Moreover, the TEM image and corresponding SAED pattern observed from the top view of the nanoplate indicate a single-crystalline structure with a spinel $Fd\bar{3}m$ space group (Fig. 3d, e and Supplementary Fig. S13). When further viewing high-resolution TEM of four positions in the sample, FT patterns exhibit completely the same signals with (002) and (220) facets (Fig. 3f), further confirming SC-MHEO is single-crystalline with (110) exposed facet. Successful synthesis of SC-MHEO is also confirmed by high-resolution XPS of metal species (Supplementary Fig. S14). Compared with SC-M-$Co_3O_4$, Co 2$p$ XPS spectra of SC-MHEO show a negative shift of 0.40 eV, indicating that alloying other metal atoms adjusts the electronic structure of Co sites. More importantly, in the Co XPS spectra of SC-MHEO, the ratio of $Co^{3+}/Co^{2+}$ increases from 1.82 to 2.86. This shows that $Mn^{2+}$, $Cu^{2+}$ and $Zn^{2+}$ metal ions substitute more $Co^{2+}$ sites, resulting in an increase in $Co^{3+}$ species. In the oxidation reaction, more adjustable $Co^{3+}/Co^{2+}$ and $Ni^{3+}/Ni^{2+}$ valence states are beneficial for electrocatalysis.

The above-detailed characterizations corroborate the successful synthesis of high-quality two-dimensional SC-MHEO nanoplates with uniform and homogeneous structural and crystalline features. As far as we are aware, low-dimensional and single-crystalline HEOs have never been prepared and reported in the literature. We deduce that the formation of SC-MHEO nanoplates is the result of careful control over the high-temperature treatment of two-dimensional SC-HEBCSs by a BCS-oxide transition route. There are some $OH^-$ and $CO_3^{2-}$ of BCS-$(CoNiMnCuZn)_2(OH)_2CO_3$. During the high-temperature treatment, both $H_2O$ and $CO_2$ are released accordingly, which thus self-template the formation of abundant penetrated mesopores. The nonporous structure of BCS-$(CoNiMnCuZn)_2(OH)_2CO_3$ with a low BET surface area of 9.7 m$^2$ g$^{-1}$ is also confirmed by $N_2$ sorption isotherms (Supplementary Fig. S15). A minor volume shrinkage also confirms this released process. Meanwhile, the smooth and controlled conversion of single-crystalline $(CoNiMnCuZn)_2(OH)_2CO_3$ does not change the crystallinity and morphology, resulting in the in situ conversion and synthesis of two-dimensional SC-MHEO-$(CoNiMnCuZn)_3O_4$ with single-crystalline and nanoplate structure. This formation mechanism is similar to the dealloying synthesis of mesoporous metals[63–65].

Our BCS-oxide conversion route is synthetically facile and general; it can be easily applicable to the preparation of other two-dimensional SC-MHEO nanoplates with different metal compositions. First, we easily change the kinds of metal precursors and form different two-dimensional SC-HEBCSs (Supplementary Figs. S16–S19). After the high-temperature conversion under the same condition, SC-MHEO nanoplates with different compositional functions are prepared accordingly (Fig. 4a). Here, three kinds of $Co_3O_4$-like SC-MHEO nanoplates, including quinary $(CoNiMnCuFe)_3O_4$, senary $(CoNiMnCuZnBi)_3O_4$, and septenary $(CoNiMnCuZnFeBi)_3O_4$, are synthesized as the typical examples. Structural characterizations clearly reveal that all the products are morphologically two-dimensional nanoplate, structurally mesoporous, and crystallographically single-crystalline and spinel (Fig. 4b–d). Furthermore, by changing the treatment atmosphere from air to $N_2$, the oxidation state of metal in $M_2(OH)_2CO_3$ cannot be further oxidized and thus remains +2. As the calcination temperature increases, $M_2(OH)_2CO_3$ structure gradually transforms into two-dimensional CoO-like SC-MHEO-$(CoNiMnCuZn)O$ nanoplates with a $Fm\bar{3}m$ space group (PDF#97-000-9865) (Fig. 4e, and Supplementary Figs. S19, S20). After the calculation, the configurational entropies ($S_{config}$) of SC-MHEO are >1.5 R, further indicating they

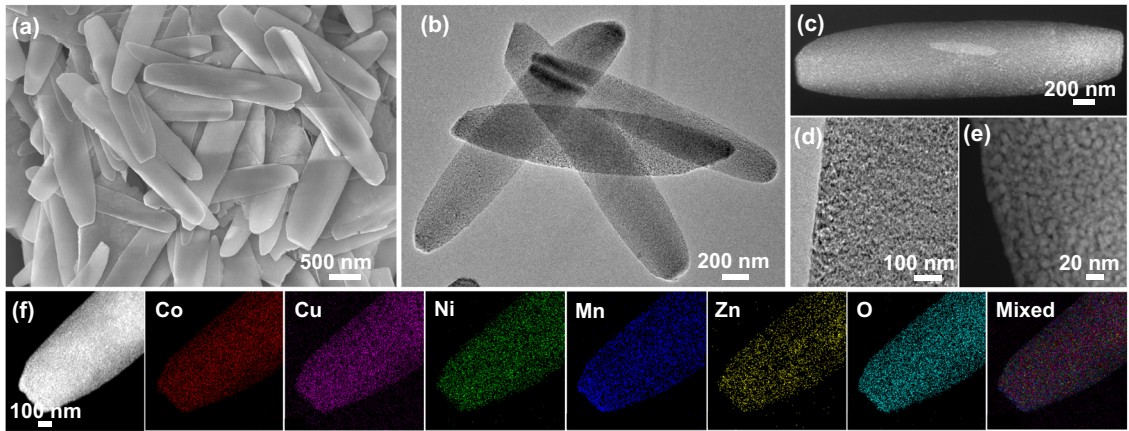

**Fig. 2 | Mesoscopic characterizations. a** Low-magnification SEM image, **b** TEM and **c** HAADF-STEM images, **d** high-magnification TEM and **e** HAADF-STEM images, and **f** HAADF-STEM EDS mapping images of SC-MHEO.

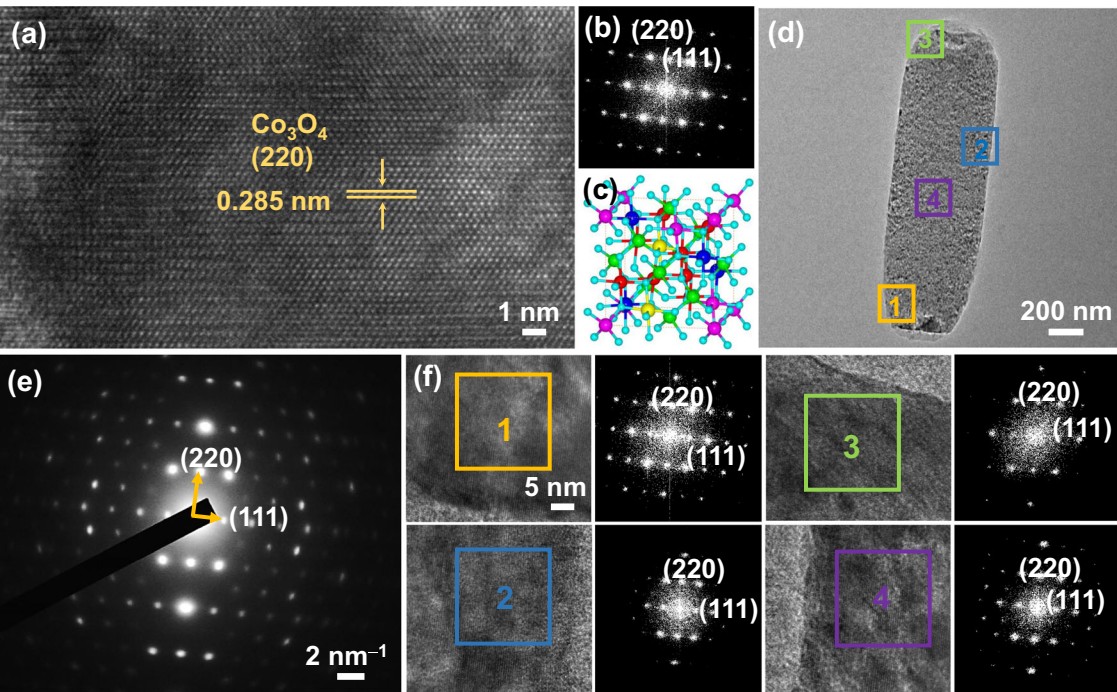

**Fig. 3 | Atomic characterizations. a** High-resolution TEM image, and **b** corresponding FT pattern and **c** structural model images of SC-MHEO. **d** TEM image and **e** corresponding SAED pattern, and **f** high-resolution TEM images collected from different regions in (**d**) and corresponding FT patterns of SC-MHEO.

are high-entropy materials (Supplementary Table S3). Meanwhile, both single-crystalline and mesoporous structures are maintained well, further confirming the generality of our BCS-oxide conversion route to the extended synthesis of a library of two-dimensional SC-MHEO nanoplates with rationally controlled compositional functions and phase structures (Supplementary Fig. S21).

Selective HMFOR electrocatalysis of SC-MHEO-(CoNiMnCuZn)$_3$O$_4$ for FDCA electrosynthesis is then performed in 1.0 M KOH containing 10 mM HMF. Meanwhile, SC-M-Co$_3$O$_4$ and P-HEO-(CoNiMnCuZn)$_3$O$_4$ are also tested as its counterpart catalysts for sharp comparisons. Compared to the linear sweep voltammetry (LSV) curve collected in the absence of HMF, all current densities electrocatalyzed by SC-MHEO are negatively shifted toward the lower potentials in the presence of HMF (Fig. 5a). Remarkably, the current density of 10 mA cm$^{-2}$ in the presence of HMF is only 1.43 V, which is 0.16 V negative than that in the absence of HMF, indicating the high electrocatalytic activity of SC-MHEO for HMFOR electrocatalysis while inhibiting its competitive

OER. Meanwhile, we compare the HMFOR performance of SC-MHEO, P-HEO, and SC-M-Co$_3$O$_4$. As shown in Fig. 5b, carbon paper (CP) as the electrocatalyst support is almost inactive for HMFOR. By contrast, both SC-MHEO and P-HEO hold a similar onset potential of 1.18 V, which is 0.17 V lower than that of SC-M-Co$_3$O$_4$ (1.35 V), indicating the high-entropy function for promoting HMFOR electrocatalysis. Moreover, SC-MHEO exhibits a higher current density than P-HEO at the potentials above 1.45 V, suggesting the importance of a single-crystalline and mesoporous structure that synergistically boosts HMFOR electrocatalysis. The reaction rates and kinetics are further evaluated by summarizing the Tafel slopes (Fig. 5c). Obviously, SC-MHEO discloses the lowest Tafel slope of 191.4 mV dec$^{-1}$, which is lower than that of P-HEO (258.6 mV dec$^{-1}$) and SC-M-Co$_3$O$_4$ (388.1 mV dec$^{-1}$), indicating SC-MHEO with single-crystalline/mesoporous structures and high-entropy compositions accelerates the reaction kinetics and thus promotes HMFOR electrocatalysis. The accelerated kinetics of SC-MHEO is also confirmed by the smallest impedance arc diameter in

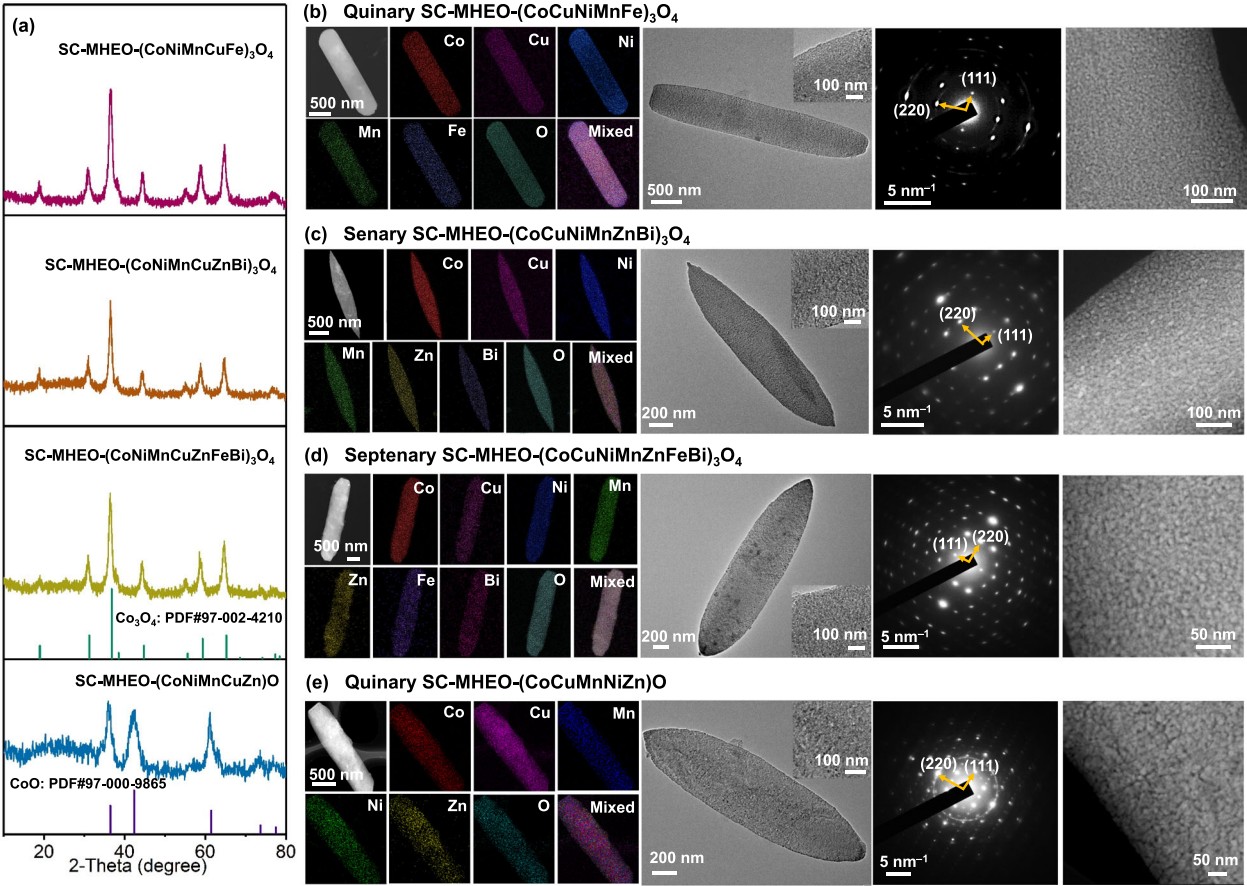

**Fig. 4 | Synthetic methodology. a** Powder XRD patterns of SC-MEHOs with different compositions and phases. HAADF-STEM EDS mapping images, TEM images and corresponding SAED patterns, and high-magnification HAADF-STEM images of **b** SC-MHEO-(CoNiMnCuFe)$_3$O$_4$, **c** SC-MHEO-(CoNiMnCuZnBi)$_3$O$_4$, **d** SC-MHEO-(CoNiMnCuZnFeBi)$_3$O$_4$, and **e** SC-MHEO-(CoNiMnCuZn)O.

electrochemical impedance spectroscopy (EIS) analysis (Supplementary Fig. S22). The double-layer capacitance (C$_{dl}$) of electrocatalysts is also tested (Supplementary Fig. S23). As summarized in Fig. 5d, SC-MHEO possesses the higher C$_{dl}$ value of 26.4 mF cm$^{-2}$, which is 24.9 and 1.7 folds higher than that of SC-M·CO$_3$O$_4$ (1.06 mF cm$^{-2}$) and P-HEO (15.8 mF cm$^{-2}$). The results indicate that SC-MHEO exposes more active metal sites for promoting HMFOR electrocatalysis. ECSA normalization of current densities of SC-MHEO and P-HEO is also compared in 50 mM HMF, indicating mesoporous structure exposes more under-coordinated metal and oxygen sites that further promote HMFOR electrocatalysis (Supplementary Fig. S24).

Chronoamperometry measurements are further performed in 1.0 M KOH and 10 mM HMF at different potentials to quantitatively identify the products of selective HMFOR electrocatalysis by high-performance liquid chromatography (Supplementary Fig. S25). As summarized in Fig. 5e, SC-MHEO discloses the high HMF conversion and FDCA selectivity in the potential range of 1.385 to 1.585 V. Typically, at lower potentials (1.385 and 1.435 V), SC-MHEO achieves extremely high HMF conversion of >97% and high FDCA Faradaic efficiency (FE) of >95%. As the potential increased further (1.485, 1.535, and 1.585 V), the FE of FDCA and the conversion rate of HMF slightly decreased, mostly because of the occurrence of competitive oxygen evolution reaction (OER). Especially at a potential of 1.435 V, the HMF conversion electrocatalyzed by SC-MHEO is as high as 99.3%, with a superior FDCA FE of 97.7%. In sharp comparisons, P-HEO and SC-M-Co$_3$O$_4$ show a relatively slower HMF conversion rate and lower FDCA selectivity at all the potentials (Supplementary Fig. S26). For example, at 1.435 V, HMF conversion and FDCA FE are 87.2% and 84.9% for P-HEO

and 78.8% and 60.5% for SC-M·Co$_3$O$_4$ (Supplementary Table S4). The results clearly demonstrate that SC-MHEO not only remarkably promotes HMFOR electrocatalysis but also dramatically enhances FDCA selectivity. At the same time, we have also conducted electrocatalytic HMFOR tests on other SC-MHEO nanoplates at the optimal voltage of 1.435 V (vs. RHE). Remarkably, all Co$_3$O$_4$-like electrocatalysts exhibited considerable conversion rates and selectivities due to the unique 'cocktail' effect similar to the performance on SC-MHEO-(CoNiMnCuZn)$_3$O$_4$ (Supplementary Fig. S27). In comparison, SC-MHEO-(CoNiMnCuZn)O disclosed the decreased activity in HMFOR electrocatalysis, which can be attributed to the absence of valence-changing metal ions in the crystal structure. Moreover, SC-MHEO shows remarkable operation stability in selective HMFOR electrocatalysis for FDCA electrosynthesis. After being tested for ten successive cycles, no significant decay is observed in both HMF conversion and FDCA selectivity (Fig. 5f). Meanwhile, SC-MHEO also maintains well in two-dimensional morphology, penetrated meso-pores, and single-crystalline structure (Supplementary Fig. S28). The results clearly highlight the synergies of high-entropy effect and single-crystalline/mesoporous structures in promoting selective HMFOR electrocatalysis (Fig. 5g). Compared with the state-of-the-art electrocatalysts reported in the literature, more impressively, SC-MHEO represents one of the most active and selective HMFOR electrocatalysts for FDCA electrosynthesis (Fig. 5h and Supplementary Table S5).

In general, there are two main reaction pathways in selective HMFOR electrocatalysis (Fig. 6a)[9,66,67]. Pathway I is the preferential electrooxidation of the aldehyde group of HMF into 5-hydroxymethyl-

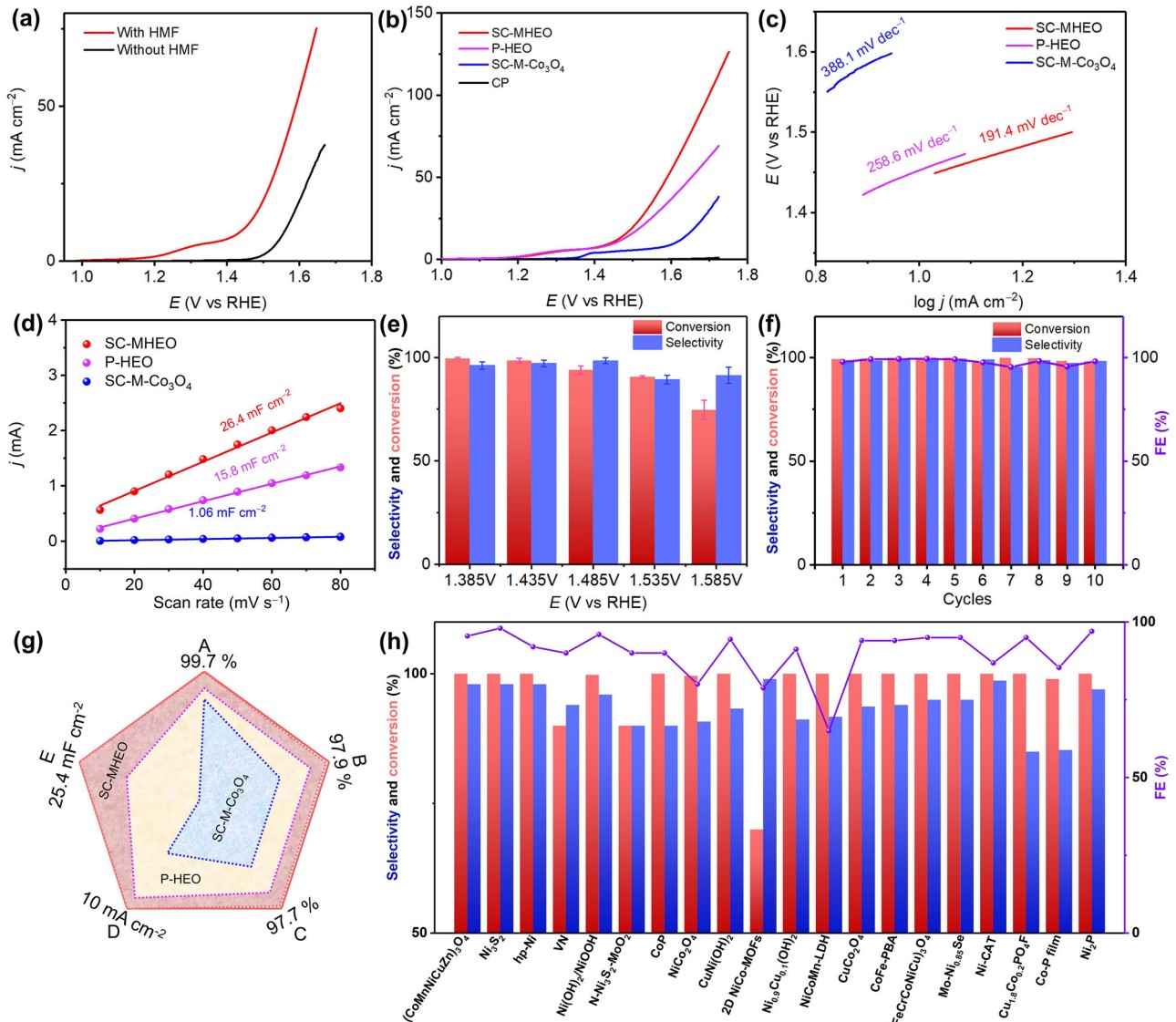

**Fig. 5 | Electrocatalytic performance. a** LSV curves of SC-MHEO collected in 1.0 M KOH with and without 10 mM HMF. **b** LSV curves and **c** summarized Tafel slopes, and **d** capacitive currents of SC-MHEO, SC-M-Co$_3$O$_4$, and P-HEO collected in 1.0 M KOH and 10 mM HMF. **e** Conversion of HMF and selectivity of FDCA for HMFOR electrocatalyzed by SC-MHEO (error bars are determined from five replicate trials at different potentials). **f** Conversion of HMF, selectivity of FDCA, and FE of FDCA electrocatalyzed by SC-MHEO for 10 consecutive cycles of HMFOR. **g** Comparisons of key HMFOR performance parameters for SC-MHEO, P-HEO, and SC-M-Co$_3$O$_4$ (A: conversion; B: FDCA yield; C: FE; D: current density at 1.43 V; E: ECSA). **h** Performance comparisons of SC-MHEO with the state-of-the-art electrocatalysts for FDCA electrosynthesis from HMFOR.

2-furancarboxylic acid (HMFCA) first (2e$^-$ route). After that, HMFCA is further oxidized into formyl-2-furancarboxylic acid (FFCA) (2e$^-$ route) and finally oxidized into FDCA (2e$^-$ reaction). In Pathway II, by contrast, HMF is first electrooxidized preferentially into 2,5-diformylfuran (DFF) by the oxidation of the hydroxyl group of HMF (2e$^-$ route). DFF is then oxidized into FFCA and finally into FDCA[68]. To probe the reaction pathway of our electrocatalysts, we perform potentiostatic electrocatalysis at 1.435 V and further analyze the concentrations of reactants, intermediates, and products. During HMFOR electrocatalysis, there is no DFF detected for SC-MHEO at different coulombic charges. Alternatively, trace of HMFCA is observed as the key reaction intermediate at the same time, indicating Pathway I dominates in our electrocatalyst (Fig. 6b and Supplementary Fig. S29)[69–71]. In sharp comparisons, more HMFCA amounts are detected for P-HEO and SC-M-Co$_3$O$_4$ in different potentials (Fig. 6c, and Supplementary Figs. S30, S31), further confirming that Pathway I dominates for HMFOR electrocatalysis. Remarkably, SC-MHEO nanoplates with high-entropy and structural

advantages synergistically accelerate further HMFCA electrooxidation and thus promote HMFOR electrocatalysis via Pathway I[8,72]. Then, we perform HMFCA electrocatalysis to highlight the high performance of SC-MHEO in the electrooxidation of the hydroxyl group in intermediate HMFCA. As summarized in Fig. 6d, SC-MHEO completely electrooxidizes HMFCA into FDCA with a superior selectivity of >99%. However, in the same test conditions, 9.5% and 42% of HMFCA are retained when electrocatalyzed by P-HEO and SC-M-Co$_3$O$_4$, respectively (Supplementary Fig. S32). The results further indicate the high-entropy and structural synergies in promoting complete electrooxidation of HMF into FDCA.

Density functional theory (DFT) calculations are also conducted to reveal the intrinsic nature of the high HMFOR performance of SC-MHEO for FDCA electrosynthesis. Here, SC-MHEO is structurally simulated by random substitutions of Ni, Mn, Cu, and Zn in Co sites of spinel Co$_3$O$_4$, as characterized above (Supplementary Fig. S33). The adsorption energy of the substrate molecule (HMF) was the activity

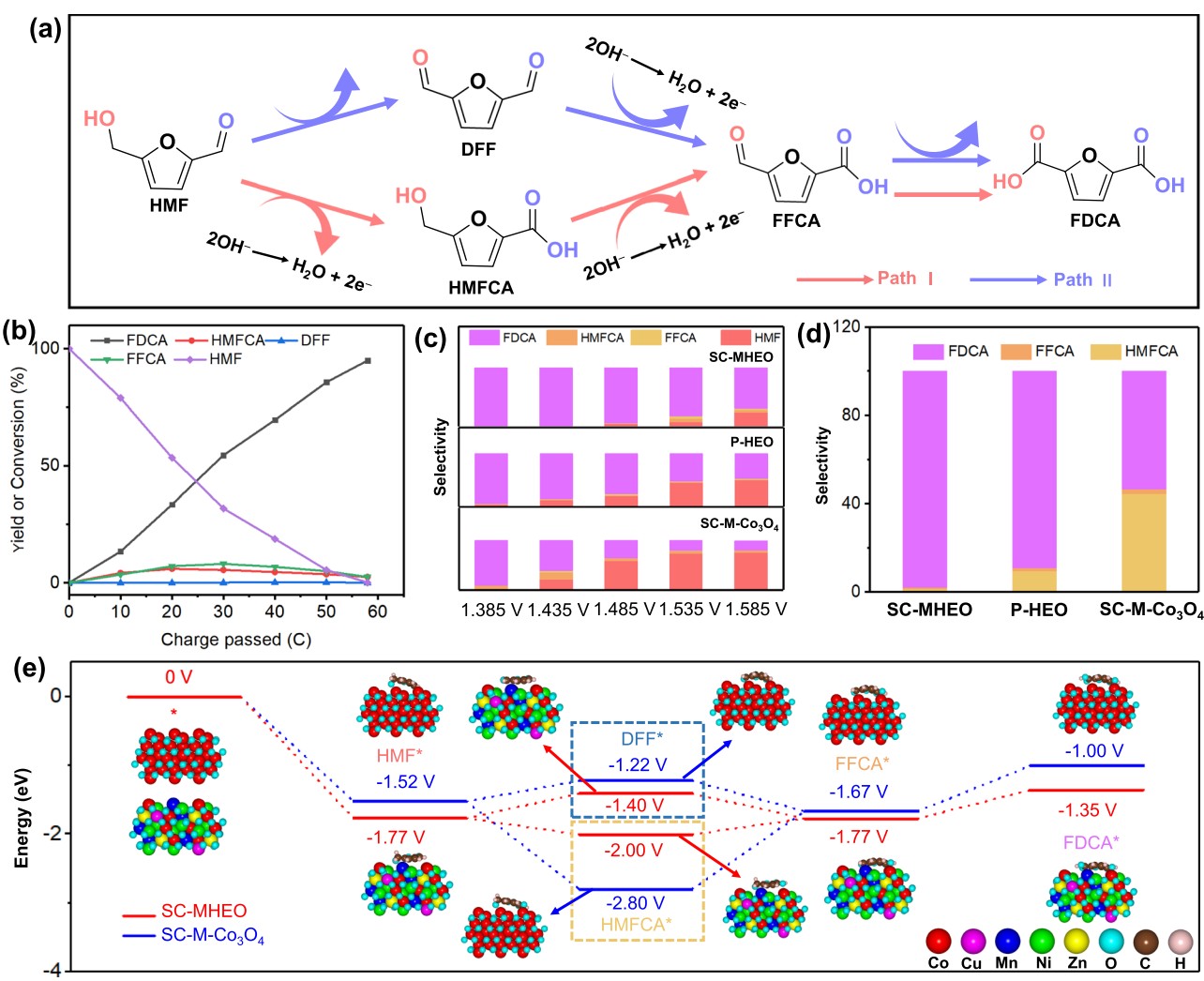

**Fig. 6 | Electrocatalytic mechanism. a** Reaction pathways of HMFOR electrocatalysis to FDCA. **b** Product distributions at 1.435 V (vs. RHE) over SC-MHEO collected in 1.0 M KOH and 10 mM HMF. **c** Product distributions at different potentials over SC-MHEO, SC-M-Co₃O₄, and P-HEO collected in 1.0 M KOH and 10 mM HMF. **d** Product distributions at 1.435 V (vs. RHE) over SC-MHEO, SC-M-Co₃O₄, and P-HEO collected in 1.0 M KOH and 10 mM HMFCA. **e** Free energies of HMFOR electrocatalysis via Pathway I and Pathway II by SC-MHEO and SC-M-Co₃O₄.

descriptor of electrochemical HMFOR. We thus calculated the adsorption energies of HMF substrate molecule on different metal sites of SC-MHEO. As presented in Supplementary Fig. S34, the Co site has the strongest adsorption of HMF (−1.77 eV), which thus was considered the main active site for HMFOR electrocatalysis. In addition, we also calculated the *d*-band center of metal elements in SC-MHEO. The results showed that the *d*-band center of Co (−2.45 eV) was larger than that of other metals (Ni: 4.92 eV, Cu: 3.11 eV, Mn: 5.07 eV, Zn: 6.44 eV) closer to the Fermi level (Supplementary Fig. S35). It showed that the Co site had a strong ability to capture reaction intermediates, which further supported our conclusion. We further calculate the energy profiles of two HMFOR pathways for SC-MHEO-(CoNiMnCuZn)₃O₄ and SC-M-Co₃O₄. As summarized in Fig. 6e, both SC-MHEO and SC-M-Co₃O₄ show the increased energy barriers of 0.37 eV and 0.30 eV, respectively, for selective electrooxidation of the hydroxyl group of HMF* into DFF* by Pathway II. Obviously, they are thermodynamically unfavorable. In comparison, in Pathway I, energy barriers of HMF*-to-HMFCA* decrease for both SC-MHEO and SC-M-Co₃O₄, indicating the spontaneous process for electrooxidation of aldehyde group of HMF (Supplementary Fig. S36). Therefore, HMFOR electrocatalysis is energetically proceeded by Pathway I, as experimentally confirmed above[73]. Moreover, the highest energy barrier of SC-M-Co₃O₄ reaches

1.13 eV for the selective HMFCA*-to-FFCA* route, indicating that it is the rate-determining step of HMFOR electrocatalysis. It strongly corresponds to the high concentration of HMFCA in the reaction. By contrast, the rate-determining step of SC-MHEO is the FFCA*-to-FDCA* route, since it needs to overcome the highest energy barrier of 0.42 eV during the electrocatalysis. Remarkably, the lower energy barrier in the rate-determining step of SC-MHEO further highlights the high-entropy effect in promoting selective HMFOR electrocatalysis, and thus results in higher activity and selectivity simultaneously.

In addition, the *d*-band center of Co in SC-MHEO is closer to the Fermi level than that of SC-M-Co₃O₄, indicating a strong ability to capture reaction intermediates, which is consistent with our reaction pathway diagram. Under the influence of other elements, the peak patterns of Co partial projected density of states (PDOS) tend to be more numerous and broader in the SC-MHEO structure than the SC-M-Co₃O₄ structure. This illustrates the impact of high-entropy systems on the central electronic structure (Supplementary Fig. S37). This illustrates the impact of high-entropy oxides on the central electronic structure. In addition, HMFOR and oxygen evolution reaction (OER) are two competing reactions due to possible water oxidation side reactions in aqueous solution. Gibbs free energies of two pathways, including the 4-electron oxygen evolution reaction and the 2-electron

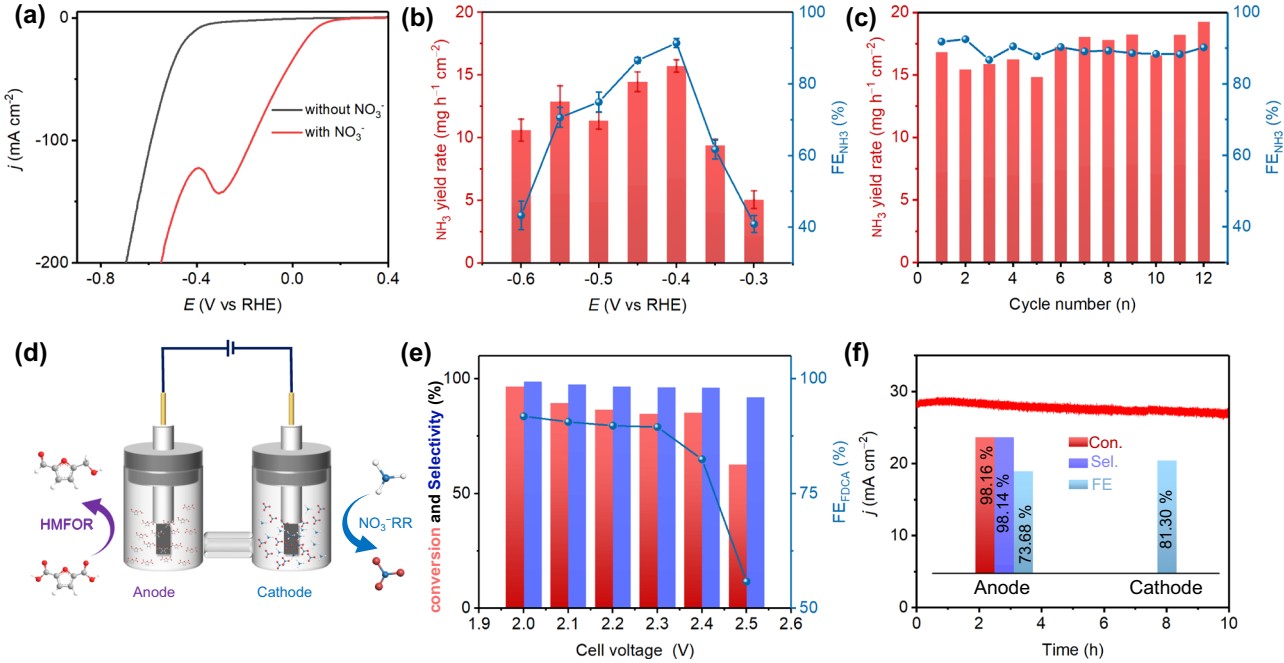

**Fig. 7 | Two-electrode cell. a** LSV curves of SC-MHEO collected in 1.0 M KOH with and without 0.10 M KNO$_3$. **b** NH$_3$ yield rates and FE$_{NH3}$ values of SC-MHEO collected in 1.0 M KOH and 0.10 M KNO$_3$ (error bars are determined from five replicate trials at different potentials). **c** Recycling stability tests of SC-MHEO collected in 1.0 M KOH and 0.10 M KNO$_3$. **d** Schematic illustration of two-electrode coupling system for anode HMFOR and cathode NO$_3^-$RR electrocatalysis. **e** Conversion of HMF and selectivity of FDCA for HMFOR electrocatalysis by SC-MHEO in a two-electrode coupling system. **f** Conversion, selectivity and FE of cathode and anode in the two-electrode cell after being tested for 10 h.

water oxidation reaction, are calculated accordingly (Supplementary Fig. S38). The results show that the reaction energy of the first step of both reactions (H$_2$O → OH* + H+ +e$^-$) reaches +1.54 eV, which is much higher than any step in the reaction pathway in HMFOR electrocatalysis. Therefore, the two side reactions do not occur preferentially.

Electrocatalytic NO$_3^-$RR is then performed in 1.0 M KOH containing 0.10 M NO$_3^-$ as a potential coupling cathode reaction of anode HMFOR electrocatalysis. LSV curves show that the current density of SC-MHEO electrocatalyst increases sharply in the presence of KNO$_3$, indicating its high activity for NO$_3^-$RR electrocatalysis (Fig. 7a). Chronoamperometry measurements are further performed at different potentials to identify the products of selective NO$_3^-$RR electrocatalysis. Considering the main product of NH$_3$, we here determine and analyze NH$_3$ produced by a typical colorimetric method (Supplementary Fig. S39). The origin of produced NH$_3$ was identified through the $^{15}$N isotope labeling experiments. The typical $^{15}$NH$_3$ peak can be seen when using $^{15}$NO$_3^-$ as nitrogen source, indicating that the NH$_3$ produced comes from NO$_3^-$RR (Supplementary Fig. S40). As summarized in Fig. 7b, both the NH$_3$ yield rate and FE$_{NH3}$ of SC-MHEO show the typical volcanic trends in the potential ranging from −0.30 to −0.60 V. Specifically, SC-MHEO discloses the best NO$_3^-$-to-NH$_3$ performance at −0.40 V with the highest FE$_{NH3}$ of 91.5% and NH$_3$ yield rate of 15.73 mg h$^{-1}$ cm$^{-2}$. In addition, SC-MHEO also shows excellent electrocatalytic NO$_3^-$-to-NH$_3$ stability. After being performed for 12 consecutive cycles at −0.40 V, there is almost no decreasing trend of SC-MHEO in NH$_3$ yield rates and FE$_{NH3}$ for NO$_3^-$-to-NH$_3$ electrocatalysis (Fig. 7c). Physical characterizations also show that structure and crystallinity of SC-MHEO catalyst retain well (Supplementary Fig. S41). These results clearly confirm that NO$_3^-$ can be efficiently electroreduced into value-added NH$_3$ with high activity and selectivity at the cathode.

Inspired by the excellent electrocatalytic performance, we finally construct a two-electrode coupling system in an H-type cell that includes selective HMF-to-FDCA electrocatalysis as the anode reaction and selective NO$_3^-$-to-NH$_3$ electrocatalysis as the cathode reaction with SC-MHEO as a bifunctional electrocatalyst (Fig. 7d). In comparison to (+) HMFOR||HER (−) system, (+) HMFOR||NO$_3^-$RR (−) coupling system shows the lower onset potentials, indicating the high potential of bifunctional SC-MHEO electrocatalyst in two-electrode coupling system (Supplementary Fig. S42). Specifically, at the current density of 10 mA cm$^{-2}$, the cell voltage of (+) HMFOR||NO$_3^-$RR (−) coupling system is as low as 1.69 V. We further summarize the product selectivity and FE of two-electrode coupling system at different cell voltages. For anode HMF-to-FDCA electrocatalysis, SC-MHEO discloses superior FDCA selectivity of >95% and high FDCA FE of >90% at the lower voltages (<2.3 V) (Fig. 7e). Specifically, the best HMF-to-FDCA performance, including 96.7% of HMF conversion rate, 98.8% of FDCA selectivity, and 91.8% of FDCA FE, is achieved at the coupling voltage of 2.0 V. For cathode NO$_3^-$-to-NH$_3$ electrocatalysis, high FE$_{NH3}$ of >80% is also achieved (Supplementary Fig. S43). The stability of the two-electrode coupling system is further characterized by electrolyzing a large volume of solution for a long time at a battery voltage of 2.2 V. Chronoamperometry test shows that, after being evaluated for 10 h continuous electrocatalysis, FDCA selectivity and FE at the anode still reach 98.1% and 73.7% with an HMF conversion of 98.2% (Fig. 7f). At the same time, FE$_{NH3}$ at the cathode is 81.3% in the same electrocatalytic condition. The results highlight that co-electrocatalysis by replacing HER with NO$_3^-$RR not only produces more value-added product (NH$_3$) but also enhances energy efficiency in the two-electrode system, thereby demonstrating their great potential for practical application. In addition, a two-electrode coupling system continuous flow electrolyzer was used to evaluate the practicality of SC-MHEO cathode for HMFOR electrocatalysis (Supplementary Fig. S44). Impressively, the SC-MHEO electrocatalyst discloses a superior selectivity of 98.3%, a high FDCA yield of 87.5%, and a remarkable FE of 86.1% in a continuous flow electrolyzer. The result further highlights the potential application of SC-MHEO in real flow electrolyzer for producing high-value-added chemicals.

## Discussion

We have successfully prepared a novel series of two-dimensional SC-MHEO nanoplates with single-crystalline and penetrated mesoporosity by a template-free synthetic strategy. This method relies on the direct conversion of parent HEBCS nanoplates under high temperature in the absence of any templates, where the released $H_2O$ and $CO_2$ produce abundant and penetrated mesopores and retain single-crystalline and two-dimensional nanostructure. With this strategy, four $Co_3O_4$-like phase SC-MHEOs (quinary $(CoNiMnCuZn)_3O_4$ and $(CoNiMnCuFe)_3O_4$, senary $(CoNiMnCuZnBi)_3O_4$, and septenary $(CoNiMnCuZnFeBi)_3O_4$) and one CoO-like phase SC-MHEO $((CoNiMnCuZn)O)$ are obtained with well-defined two-dimensional plate-like nanostructure. Our SC-MHEOs feature multiple structural advantages, including high-entropy composition, mesoporous structure and single-crystalline phase, holding a remarkable electrocatalytic performance for efficient biomass upgrading. Typically, the best SC-MHEO-$(CoNiMnCuZn)_3O_4$ nanoplates disclose a remarkable HMF conversion of 99.3%, a superior 2,5-furandicarboxylic acid (FDCA) selectivity of 97.7%, and a high cycling stability (almost no delay for 10 cycles). Mechanism studies ascribe the high performance of SC-MHEO to the high-entropy and single-crystalline/mesoporous structures that not only expose more electrocatalytically active sites but also optimize energy barriers (and the rate-determining steps) of HMFOR. More importantly, SC-MHEO behaves as an efficient bifunctional electrocatalyst in the two-electrode coupling system that not only produces more value-added products but also enhances energy efficiency.

Our study has offered two important implications for both the discovery of new materials and the exploration of new applications. On the one hand, our BSC template transition strategy can be easily extended to prepare other high-entropy materials with controlled compositional functions, including alloys, sulfides, phosphides, and nitrides. For example, the BSC-phosphide transition route can be performed in the presence of $NaH_2PO_2$ that thus prepares high-entropy mesoporous phosphides. These materials would be potentially applied as new materials for various applications. On the other hand, our findings offer an alternative strategy by contrasting the two-electrode coupling system that concurrently achieves green and energy-efficient value-added products from renewable non-fossil biomass-derived chemicals and other wastes. Considering abundant anode/cathode reactions in water, some new applications that couple different oxidation and reduction reactions into value-added products are highly desired.

## Methods

### Synthesis of SC-MHEO

Typically, 8.5 mmol of urea was dissolved in 10 mL of deionized water to obtain a homogeneous solution, followed by the addition of 15 mL of ethanol. Then, Solution A was formed after the addition of 4.0 mL of oleic acid and stirred for 30 min. Solution B was prepared by adding 0.60 mmol of cobalt nitrate, 0.50 mmol of nickel nitrate, 0.30 mmol of manganese nitrate, 0.20 mmol of copper nitrate and 0.20 mmol of zinc nitrate to 10 mL of deionized water. After that, Solution A was mixed with Solution B and further stirred for 1 h. Subsequently, the above solution was transferred to a polytetrafluoroethylene-lined stainless steel autoclave and reacted at 160 °C for 9 h. After being cooled to room temperature, the product was collected by sequentially washing with n-hexane, ethanol, and water several times and further drying at 60 °C. As-obtained $(CoNiMnCuZn)_2(OH)_2CO_3$ was finally calcined in air at 300 °C for 3 h to obtain SC-MHEO-$(CoNiMnCuZn)_3O_4$. Extended syntheses of other SC-MHEOs and P-HEO are presented in detail in Supplementary Information.

### Calculation of $S_{config}$

$S_{config}$ was calculated according to the following Eq. (1)

$$S_{config} = -R \left[ \left( \sum_{a=1}^{n} x_a \ln x_a \right)_{cation-site} + \left( \sum_{b=1}^{m} x_b \ln x_b \right)_{anion-site} \right] \tag{1}$$

n and m are the number of element types, $x_a$ and $x_b$ are the mole fractions of element components at cation sites and anion sites, and R is the universal gas constant ($R = 8.314\,J\,K^{-1}\,mol^{-1}$). The molar fraction used for entropy calculation was obtained from the ICP results. Materials with $S_{config} \geq 1.5\,R$ are considered high-entropy systems, while materials with $1.5\,R > S_{config} \geq 1\,R$ and $S_{config} < 1\,R$ are considered medium-entropy and low-entropy systems, respectively.

### Electrocatalytic HMFOR tests

All electrochemical experiments were performed at room temperature (25 °C) using a CHI Instruments electrochemical analyzer (CHI 760E) in an H-type electrolytic cell. A standard three-electrode system, consisting of a Pt pillar electrode as the counter electrode, saturated Hg/HgO as the reference electrode, and a catalyst-loaded carbon paper electrode as the working electrode, was used for electrochemical experiments. CV measurements were performed in 1.0 M KOH solution (scan rate 50 mV s$^{-1}$). LSV measurements were performed in 1.0 M KOH solution (scan rate: 10 mV s$^{-1}$). All LSV curves are without IR compensation. Electrochemical impedance spectroscopy (EIS) tests were measured over a frequency range from $10^{-1}$ to $10^5$ Hz. CV curves in electrochemical double-layer capacitance ($C_{dl}$) determinations were measured in a potential window nearly without the Faradaic process at different scan rates of 10, 20, 30, 40, 50, 60, 70, and 80 mV s$^{-1}$. The plot of current density at set potential against scan rate has a linear relationship, and its slope is the $C_{dl}$.

### Electrocatalytic $NO_3^-$RR tests

All $NO_3^-$RR experiments were performed using a three-electrode system, in which Hg/HgO and a platinum column served as the reference and counter electrodes, respectively. A solution containing 0.1 M $NO_3^-$ ($KNO_3$) and 1 M KOH is used as the electrolyte, and the working environment of the electrolytic cell is maintained in an Ar gas (ultra-high purity, 99.999%) atmosphere.

### Determination of ammonia

The amount of $NH_3$ produced in the reaction solution was determined by a colorimetric method. First, 2.0 mL of dilute electrolyte was mixed with 2.0 mL of 1.0 M NaOH solution containing salicylic acid and sodium citrate. Then, 1.0 mL of 0.05 M sodium hypochlorite and 0.20 mL of 1.0 wt% sodium nitroferricyanide dihydrate were added to the above solution. After standing at room temperature in the dark for 1 h, the absorbance at a wavelength of 655 nm in the UV-vis absorption spectrum was used to measure the concentration of $NH_3$ produced.

### Electrochemical measurements of (+) HMFOR||$NO_3^-$RR (−) coupled system

The anode cell consisted of a 1.0 M KOH with 10 mM HMF solution and a working electrode dripped with 2.0 mg of catalyst ink. The cathode cell consists of a 1.0 M KOH with 0.10 M $KNO_3$ solution and a working electrode dripped with 1.0 mg of catalyst ink. Electrolyze a certain amount of charge and collect the electrolyte from the cathode and anode respectively for product analysis.

### Flow electrolyzer

The continuous flow electrolyzer anolyte consisted of 100 mM HMF in 100 mL of 1.0 M KOH. Using a peristaltic pump to pump the

electrolyte into the reaction system. The flow electrolyzer has an electrode area of 9.0 cm² and uses a proton exchange membrane to separate the anode and cathode of the electrolyzer. The loading is 2.0 mg cm⁻². In addition, the electrolysis reaction was performed at room temperature with a constant current of 2.1 V. The product was confirmed by HPLC.

## Reporting summary

Further information on research design is available in the Nature Portfolio Reporting Summary linked to this article.

## Data availability

This study is available from the corresponding author upon request. The source data underlying Figs. 1c–f, 4a, 5a–f, 6b–e and 7a–c, 7e, f are provided as a Source Data file. Source data are provided with this paper.

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

## Acknowledgements

B.L. thanks the financial support from the National Key Research and Development Program of China (2023YFC3905002), the Sichuan Science and Technology Program (2023ZYD0039), and the Fundamental Research Funds for the Central Universities. H.L. acknowledges the financial support from the China Postdoctoral Innovation Talent Support Program (BX20230221). The authors would like to thank Dr. Yanping Huang (Center of Engineering Experimental Teaching, School of Chemical Engineering, Sichuan University) for her assistance with SEM imaging and Dr. Feng Yang (the Comprehensive Training Platform of the Specialized Laboratory, College of Chemistry, Sichuan University) for her assistance of TEM imaging.

## Author contributions

B.L. conceived the project and designed the experiments. Y.W. and H.L. performed the catalysts preparation and analyzed the formation mechanism. Y.W., H.H., and F.J. carried out the electrochemical tests and analyzed the data. All the authors discussed the results and co-wrote the paper.

## Competing interests

The authors declare no competing interests.
