## [Peer Review File · Nature Communications]

Two-Dimensional Single-Crystalline Mesoporous High-Entropy Oxide Nanoplates for Efficient Electrochemical Biomass UpgradingREVIEWER COMMENTS

Reviewer #1 (Remarks to the Author):

In this work, 2D single-crystalline mesoporous high-entropy oxides (SC-MHEOs) nanoplates were prepared by pyrolysis of high-entropy basic carbonate salts. The resultant SC-MHEOs showed superior performance in selective HMF oxidation reaction (HMFOR) and coupled nitrate reduction. This article offers a facile and general strategy for synthesizing single-crystalline mesoporous high-entropy materials. The materials and performance have been well characterized and the mechanism has also been discussed in detail. Therefore, I recommend this work to be accepted for publication subject to a minor revision as below.

1. Considering that two-dimensional single-crystalline high-entropy materials such as metal phosphorus trichalcogenides have been previously reported (Adv. Funct. Mater. 2023, 2312322), the uniqueness of this work should be further clarified.
2. The synthetic protocol of P-HEO should be added in the "Materials and Methods" part. The atomic ratio of the elements in P-HEO should also be given.
3. The authors stated that "by changing the treatment atmosphere from air to N₂, two-dimensional CoO-like SC-MHEO-(CoNiMnCuZn)O nanoplates with a Fm-3m space group are prepared". The impact of treatment atmosphere on the crystalline phase of SC-MHEOs should be further discussed.
4. In the LSV curve in Figure 6a, an obvious peak at -0.4 V vs RHE can be observed in the presence of KNO₃, the authors should provide an explanation for the phenomenon.
5. During the theoretical calculations, how did the authors confirm the adsorption sites for binding the reactant molecules?
6. There are some errors and typos in the manuscript. For example, "CO3O4" should be "Co3O4" (page 5) and "slef-template" should be "self-template" (page 6). Scale bar is missing in Figure 2i.

Reviewer #2 (Remarks to the Author):

In this manuscript, the authors present a mesoporous high entropy oxide which presents excellent HMF conversion, FDCA selectivity and NH₃ production. However, I don't think the current edition meet the criteria of Nat. Commun. in many aspects. Basically, this work doesn't show too much innovation from synthesis to application. Moreover, a few concerns have been raised regarding the clarity of this manuscript.

1. From the XPS results, the analysis of the influence of the valence states of each element in SC-MHEO on its electrochemical performance has not been conducted, and the impact of different elements on its electronic structure lacks in-depth research.

2. In this experiment, the SC-MHEO main sample and other multicomponent materials such as SC-MHEO-(CoNiMnCuFe)₃O₄, SC-MHEO-(CoNiMnCuZrBi)₃O₄, and SC-MHEO-(CoNiMnCuZnFeBi)₃O₄, as well as samples treated in an N₂ atmosphere, were synthesized. Their performances can be compared with that of the main sample, and the superiority of the main sample can be analyzed. If there are commercial samples available, they can also be included in the comparison.
3. I doubt the so-called 2D nanoplate structure because they look like 1D rod-like structure (spindle or rice). Authors should provide topologic TEM images or AFM result for their claim.
4. Formation mechanism of mesopores is not clear. I suggest authors should also investigate the nitrogen sorption of high-entropy basic carbonate before calcination rather than simply quote previous references.
5. The voltage conditions for impedance testing are not clearly labeled.
6. The double-layer capacitance fitting of the SC-MHEO catalyst shows some deviations from the fitting line, with some points deviating from the line. It may be necessary to adjust the scanning speed to make it align completely on a straight line.
7. The active sites in high entropy alloy oxides are mentioned multiple times in this article, but the investigation of these active sites is not clear. It is uncertain whether the active sites are located on certain elements or on the bridging sites between two elements. Additionally, whether it is possible to calculate the Gibbs free energy of each element in the catalytic process by theoretical calculation?
8. Can a Flow Cell be assembled for the HMFOR reaction to explore its commercial prospects?
9. Is there a formula to calculate the configuration entropy? Does it comply with the standard entropy value of high entropy alloys?
10. The superiority of mesoporous materials has been emphasized throughout this paper, and the existence of a large number of defects and oxygen vacancies in mesopores has been mentioned. However, the mechanism of how the pores affect its performance has not been thoroughly explored, and it is unclear whether the composition of elements inside the pores is consistent with that of the shell.
11. SC-MHEO exhibits the highest conversion rate and Faradaic efficiency at a voltage of 1.435V, but in terms of performance at 1.435V, its advantage over P-HEO is too small.
12. The FT pattern in Fig S18J shows different crystal orientation. Could authors give an explanation?

Reviewer #3 (Remarks to the Author):

This work demonstrated a strategy to synthesis two-dimensional mesoporous high-entropy transition-metal oxides by thermally converting 2D high-entropy basic carbonate salts. Comparing with the traditional polycrystalline high-entropy oxides, this structure presented by the authors demonstrates a better HMF electrocatalytic oxidation performance. While I found the synthesis strategy interesting and the relevant TEM analysis attractive, the novelty of the mesoporous and

single-crystal HEO as demonstrated by the author cannot satisfy the criterion of Nature Commun. Below are listed several comments to further improve this work:

The author presented several potentially attractive properties of their HEO structure in the introduction, e.g., 2D, single-crystalline, mesoporous... Which characteristic mostly benefits the electrochemical performance and why? The reviewer cannot quickly get the designing logic of the HEO structure.

Structural characterization: 1. P5/line 130: Please explain more about the XRD comparison between monometallic Co₃O₄ and HEO. Why can the peak shift indicate the replacement of Co by other transition-metals? 2. Line 133: the crystalline structure cannot be confirmed by Raman. 3. I post my concern on the XPS analysis: the band width of O1 peak was not consistent in SC-MHEO and P-HEO.

Electrocatalysis: 1. The author should present the ECSA-normalized current density. How will the mesoporous structure contribute to the performance, only by increasing the exposure of active sites? 2. Will the single-crystalline nature contribute to the excellent performance? 3. A thorough performance comparison with literatures should be listed in the manuscript.

General: The words should be polished extensively. Many typos can be found throughout the current paper. For example, Line 213-214; Line 228-229...

Reviewer #4 (Remarks to the Author):

This manuscript reports a very facile solid-phase synthetic strategy for preparing high-quality mesoporous high-entropy oxides (MHEOs). By the utilization of pre-synthesized high-entropy basic carbonates as the parent template, in situ conversion and release of H₂O and CO₂ result in precise synthesis of high-quality MHEOs with two-dimensional and single-crystalline structures.

Meanwhile, this strategy is universal and can be extended to prepare other two-dimensional MHEOs with controlled elemental compositions and phase structures. This should be the first work on preparing two-dimensional single-crystalline MHEOs (SC-MHEOs). In the catalysis part, the authors explore the performance of SC-MHEOs in a two-electrode coupling system. SC-MHEO-(CoNiMnCuZn)₃O₄ nanoplates exhibit excellent activity and stability in both the HMFOR reaction at the anode and the NO₃-RR reaction at the cathode.

In overall, the manuscript is well-organized and written. The conclusions are also supported well by carefully conducted experiments. I believe the work would be of interest for the broad readership of the journal and will make a significant contribution to the field of two-dimensional high-entropy electrocatalysts and beyond. In my opinion, this manuscript can be accepted for publication in Nature Communications with the following revisions.

1. The author mentioned in the manuscript "In comparison to parent basic carbonate salt, SC-MHEO becomes slightly smaller with average length, center width, and thickness of 2.9 μm, 460 nm, and 70 nm, indicating a minor volume shrinkage during the calcination." However, low-magnification SEM cannot clearly explain this problem. Structural information similar to Figure S3

needs to be supplemented.

2. HMFOR and OER are two competing reactions. The authors should provide the theoretical basis for the preferential occurrence of the HMFOR reaction.

3. The author needs to identify the source of N in NH₃ in the NO₃-RR reaction at the cathode, such as ¹⁵N isotope labeling experiments.

4. In this manuscript, the pores were produced by solid-phase high-temperature treatment for the release of H₂O and CO₂. Did the authors tune the treatment temperatures? Can pore properties (sizes) be changed by temperatures?

5. During the NO₃-RR stability test, the yield of NH₃ increased after 12 cycles of testing. What is the reason?

6. In Figure 6b, the authors mentioned that NH₃ yield of SC-MHEO showed a typical volcanic trend in the potential range from -0.30 V to -0.55 V. However, this rule is not followed at -0.50 V and -0.55 V. The author should test the NH₃ yield at more negative voltages.

Response Letter for

Manuscript Title: Two-Dimensional Single-Crystalline Mesoporous High-Entropy Oxide Nanoplates for Efficient Electrochemical Biomass Upgrading Coupling with Ammonia Electrosynthesis

Manuscript Number: NCOMMS-23-58825

Comments from the Reviewers

Reviewer 1

Overall Comments: In this work, 2D single-crystalline mesoporous high-entropy oxides (SC-MHEOs) nanoplates were prepared by pyrolysis of high-entropy basic carbonate salts. The resultant SC-MHEOs showed superior performance in selective HMF oxidation reaction (HMFOR) and coupled nitrate reduction. This article offers a facile and general strategy for synthesizing single-crystalline mesoporous high-entropy materials. The materials and performance have been well characterized and the mechanism has also been discussed in detail. Therefore, I recommend this work to be accepted for publication subject to a minor revision as below.

Response: We kindly thank the Reviewer for reading through and providing such positive comments on our manuscript.

Comment 1: *Considering that two-dimensional single-crystalline high-entropy materials such as metal phosphorus trichalcogenides have been previously reported (Adv. Funct. Mater. 2023, 2312322), the uniqueness of this work should be further clarified.*

Response: We thank the Reviewer for providing the valuable comment and important reference. We have read the recommended article and promptly updated in our revised manuscript. Compared with this reported work, the SC-MHEO reported in this work has the following unique features: 1) The difference of preparation strategies. Compared with the traditional high-temperature strategy in the reference, this manuscript reported highly crystalline polybasic carbonates (BCS, $M(OH)_2CO_3$) as the parent precursors that robustly converted into high-entropy oxides at low temperatures. Moreover, single crystalline structure was inherited from the basic carbonate structure. 2) The particularity of mesoporous single crystal structure. Li et al. mentioned that “The porous architecture is free of grain boundaries, and the fully interconnected skeletons are in single-crystalline states within the basic porous frames. Single crystals with porosities can therefore be considered to be a new kind of porous material, but they are single-crystal-like because the structural symmetry is maintained only in the skeletons and completely lost within the pores. We therefore call them porous single crystals or consider them in porous single-crystalline states to stand out with their structural features.” (*Acc. Chem. Res.* 2023, 56, 3, 374) This fully illustrates the importance of this type of material with mesoporous single crystal structure. In comparison, the reported materials were poly-crystalline. 3) Elemental tunability of high-entropy oxides. The BSC-to-oxide conversion route reported in this manuscript is synthetically simple and versatile; it can be easily applied

to prepare other two-dimensional SC-MHEO nanoplates with different metal compositions and phase structures.

In summary, compared with the work reported previously (*Adv. Funct. Mater.* 2023, 34, 2312322), this work presents a major step forward in material chemistry, not only as a new and general synthetic strategy for single-crystalline mesoporous metal oxides, but also a novel design principle for new materials discovery.

Comment 2: *The synthetic protocol of P-HEO should be added in the "Materials and Methods" part. The atomic ratio of the elements in P-HEO should also be given.*

Response: We are sorry we forgot including the synthetic protocol of P-HEO in “Materials and Methods”. We have added the corresponding content in the revised Supporting Information, as read: **“Synthesis of P-HEO.** P-HEO were prepared by the reported high-temperature methods (*Angew. Chem. Int. Ed.* 2021, 60, 20253). Typically, an alkaline medium solution with a pH of 10 was prepared by adding 25 mL of ammonia (NH₃, 25%) to 200 mL of deionized water. Subsequently, a total amount of 0.10 M metal precursors were added to the above solution. The molar ratio of elements is: Co:Mn:Ni:Cu:Zn = 4.3:3.5:2.0:1.6:1.0. The mixture was stirred continuously at room temperature for 1 h. After the filtration, the precipitate was dried at 60°C for 4 h and gently ground evenly in an agate mortar. Finally, it was calcined at 950 °C for 1 h to obtain P-HEO.”

Table S2. Element ratios of P-HEOs obtained from ICP-MS.

Element	Percentage (%)
Co	33.5
Mn	26.8
Ni	16.6
Cu	15.9
Zn	7.2

Comment 3: *The authors stated that "by changing the treatment atmosphere from air to N₂, two-dimensional CoO-like SC-MHEO-(CoNiMnCuZn)O nanoplates with a Fm-3m space group are prepared". The impact of treatment atmosphere on the crystalline phase of SC-MHEOs should be further discussed.*

Response: We thank the Reviewer for this comment. According to the Reviewer’s comment, we have included the corresponding content in the revised manuscript, as read: **“Furthermore, by changing the treatment atmosphere from air to N₂, the oxidation state of metal in M(OH)₂CO₃ cannot be further oxidized and thus remains +2. As the calcination temperature increases, M(OH)₂CO₃ structure gradually transforms into two-dimensional CoO-like SC-MHEO-(CoNiMnCuZn)O nanoplates with a Fm-3m space group (PDF#97-000-9865) (Figures 3e, and S19-20)”.**

Comment 4: *In the LSV curve in Figure 6a, an obvious peak at -0.4 V vs RHE can be observed in the presence of KNO₃, the authors should provide an explanation for the phenomenon.*

Response: We thank the Reviewer for this comment. The LSV curve shows a reduction peak at -0.31 V (vs RHE), which clearly indicates that NO_3^- can be effectively reduced. The ammonia yield at a voltage of -0.30 V is much lower than -0.40 V. Combining the NO_3^- RR reduction process in an alkaline medium, we believe that the main reaction in the reduction peak at -0.31 V (vs RHE) is: $\text{NO}_3^- + \text{H}_2\text{O} + 2\text{e}^- \rightarrow \text{NO}_2^- + 2\text{OH}^-$.

Comment 5: *During the theoretical calculations, how did the authors confirm the adsorption sites for binding the reactant molecules?*

Response: We thank the Reviewer for this important comment. For HMFOR, the adsorption energy of substrate molecules is an activity descriptor for electrochemical alcohols and aldehydes (Adv. Mater. 2024, 36, 2311698; Chem. Eng. J. 2022, 433, 133842). Based on this statement, we have calculated the adsorption energies of HMF substrate molecules on five metal sites of SC-MHEO. As summarized in the revised Figure S35, the Co site has the strongest adsorption for HMF (-1.77 eV). Therefore, it is considered as the main active site for HMFOR. In addition, we have also calculated the *d* band center of each metal in SC-MHEO. The results show that the *d* band center of Co (-2.45 eV) is closer to the Fermi level than other metals (Ni: 4.92 eV, Cu: 3.11 eV, Mn: 5.07 eV, Zn: 6.44 eV) (Figure S36). This further suggests that Co is the best adsorption site. To respond the Reviewer's comment and make our statement clearer, we have included the corresponding content in the revised manuscript, as read: "The adsorption energy of substrate molecule (HMF) was the activity descriptor of electrochemical HMFOR. We thus calculated the adsorption energies of HMF substrate molecule on different metal sites of SC-MHEO. As presented in **Figure S34**, the Co site has the strongest adsorption of HMF (-1.77 eV), which thus was considered as the main active site for HMFOR electrocatalysis. In addition, we also calculated the *d*-band center of metal elements in SC-MHEO. The results showed that the *d*-band center of Co (-2.45 eV) was larger than that of other metals (Ni: 4.92 eV, Cu: 3.11 eV, Mn: 5.07 eV, Zn: 6.44 eV) closer to the Fermi level (**Figure S35**). It showed that the Co site had a strong ability to capture reaction intermediates, which further supported our conclusion."

Figure S34. Adsorption model structure of HMF at (a) Co, (b) Cu, (c) Ni, (d) Mn, and (e) Zn sites of SC-MHEO. (f) Calculated adsorption energies of Co, Cu, Ni, Mn, and Zn sites in the SC-MHEO structure.

Figure S35. PDOSs of SC-MHEO.

Comment 6: *There are some errors and typos in the manuscript. For example, "CO₃O₄" should be "Co₃O₄" (page 5) and "slef-template" should be "self-template" (page 6). Scale bar is missing in Figure 2i.*

Response: We thank the Reviewer for pointing out these carelessness mistakes. We have revised these mistakes in the manuscript. Moreover, we have also double-checked and revised the manuscript thoroughly.

Reviewer 2

Overall Comments: In this manuscript, the authors present a mesoporous high entropy oxide which presents excellent HMF conversion, FDCA selectivity and NH₃ production. However, I don't think the current edition meet the criteria of Nat. Commun. in many aspects. Basically, this work doesn't show too much innovation from synthesis to application. Moreover, a few concerns have been raised regarding the clarity of this manuscript.

Response: We thank the Reviewer very much for reading through and commenting on our manuscript. We fell very strongly that our work presents a major step forward in materials chemistry, not only *a new and highly efficient strategy* for synthesis and discovery of new materials that disclose single-crystalline mesoporous high-entropy oxides, but also *a novel catalysts design principle* for property optimization and application exploration where the two-electrode coupling systems are used to concurrently prepare value-added products with a lower energy consumption. According to the Reviewers' comments, we have further highlighted important innovations as follow:

1) **The innovation of synthetic method.** As presented in Figure 1a, in this work, we developed a BSC-oxide transition strategy to preparing two-dimensional single-crystalline mesoporous high-entropy oxides (SC-MHEOs). This route is completely different to previous works, and can be extended to various SC-MHEOs with different metal compositions and crystalline phases. To the best of our knowledge, low-dimensional single-crystalline mesoporous HEOs with well-defined nanostructures and controlled metal compositions have never been achieved with previously reported methods thus far. Besides, our BSC template transition strategy can be easily extended to prepare other high-entropy materials with controlled compositional functions, including alloys, sulfides, phosphides, nitrides. Therefore, this type of synthetic method is universal and innovative enough.

2) **The innovation of materials discovery.** Single-crystalline mesoporous transition metal oxides contribute a new class of nanostructured materials and have behaved their wide utilizations in catalysis and electrocatalysis. As Xie et al. commented on this type of material, "Porous single crystals at the macroscale combine the advantages of porous materials and single crystals to incorporate both porosity and structural coherence in a porous architecture, leading to invaluable opportunities to alter the material's properties by controlling the unique structural features to enhance its performance" (Acc. Chem. Res. 2023, 56, 374). Despite great potential, most of mesoporous oxides reported in the literature are low-crystalline and/or polycrystalline, which potentially impeded their theoretical investigations and practical applications to some extent. In this work, we found that SC-MHEOs exposed more undercoordinated metal and oxygen sites, tunable metal valences, and "cocktail" effects, all of which thus promoted electrocatalysis. Such a material and more importantly its deviates (alloys, sulfides, phosphides, nitrides) have been reported in the literature. Therefore, these types of materials are innovative enough.

3) **The innovation of application exploration.** As the Reviewer mentioned, SC-MHEOs disclose excellent HMF conversion, FDCA selectivity, and NH₃ production. This means that these materials exhibit unusual catalytic activity and stability during the catalytic process through the high-entropy "cocktail" effect and structural advantages of mesopores and single crystals. This provides new opportunities for catalytic conversion of biomass-derived chemicals to obtain high value-added products. More impressively, with nitrate reduction as the coupling cathode reaction, SC-MHEO realizes concurrent

electrosynthesis of value-added FDCA and ammonia with a lower energy consumption in the two-electrode cell. Considering abundant anode/cathode reactions in water, other new applications that couple different oxidation and reduction reactions into value-added products are highly desired. Therefore, these application explorations are innovative enough.

On the basis of above discussion, we strongly believe our work have offered wide opportunities for materials discovery, property optimization, and application exploration. We have noted that the Reviewer 1 ranked that “*The materials and performance have been well characterized and the mechanism has also been discussed in detail*” and Reviewer 4 ranked that “*In overall, the manuscript is well-organized and written... I believe the work would be of interest for the broad readership of the journal and will make a significant contribution to the field of two-dimensional high-entropy electrocatalysts and beyond.*” We thus expect that the Reviewer 2 can agree with our statements and recommend the acceptance of our work for the publication in *Nat. Commun.* Besides, **all the response details have been presented below for the evaluation of Reviewer 2.**

Comment 1: *From the XPS results, the analysis of the influence of the valence states of each element in SC-MHEO on its electrochemical performance has not been conducted, and the impact of different elements on its electronic structure lacks in-depth research.*

Response: We thank the Reviewer for this valuable comment. We strongly agree with the Reviewer’s comment that the valence state of each element in SC-MHEO would change the electronic state and thus effect their electrochemical performance. According to the Reviewer’s comment and make our statement clearer, we have included the corresponding content in the revised manuscript, as read: “**Successful synthesis of SC-MHEO is also confirmed by high-resolution XPS of metal species (Figure S14). Compared with SC-M-Co₃O₄, the XPS spectrum of Co 2p^{3/2} shows a negative shift of 0.40 eV, indicating that alloying other metal atoms adjusts the electronic structure of Co center. More importantly, in the Co XPS spectrum of SC-MHEO, the ratio of Co³⁺/Co²⁺ increased from 1.82 to 2.86. This shows that Mn²⁺, Cu²⁺ and Zn²⁺ metal ions replace more Co²⁺ sites, resulting in an increase in Co³⁺ species. In the oxidation reaction, more adjustable Co³⁺/Co²⁺ and Ni³⁺/Ni²⁺ valence states are beneficial to the progress of the reaction.**”

Figure S14. High-resolution XPS spectra of (a) Co 2p for SC-M-Co₃O₄, high-resolution XPS spectra of (b) Co 2p, (c) Ni 2p, (d) Mn 2p, (e) Cu 2p, and (f) Zn 2p for SC-MHEO-(CoNiMnCuZn)₃O₄.

Note for Figure S14: In the Co XPS spectrum of SC-M-Co₃O₄, the Co 2p^{3/2} peaks located at 780.5 eV and 781.9 eV are attributed to Co³⁺ and Co²⁺, respectively. The ratio of Co³⁺/Co²⁺ is 1.82. In the XPS spectrum of SC-MHEO, the Ni 2p^{3/2} peaks located at 855.2 eV and 857.3 eV are attributed to Ni²⁺ and Ni³⁺, respectively. The Mn 2p^{3/2} and Mn 2p^{1/2} peaks located at 642.5 eV and 654.6 eV are attributed to Mn²⁺. The Cu 2p^{3/2} and Cu 2p^{1/2} peaks located at 934.1 eV and 953.9 eV are attributed to Cu²⁺. The Zn 2p^{3/2} and Zn 2p^{1/2} peaks located at 1021.7 eV and 1044.7 eV are attributed to Zn²⁺.

Comment 2: In this experiment, the SC-MHEO main sample and other multicomponent materials such as SC-MHEO-(CoNiMnCuFe)₃O₄, SC-MHEO-(CoNiMnCuZnBi)₃O₄, and SC-MHEO-(CoNiMnCuZnFeBi)₃O₄, as well as samples treated in an N₂ atmosphere, were synthesized. Their performances can be compared with that of the main sample, and the superiority of the main sample can be analyzed. If there are commercial samples available, they can also be included in the comparison.

Response: We thank the Reviewer for this comment. In previous version, we have extended our strategy to synthesize various SC-MHEO nanoplates with different compositions and phase structures. We thus did not discuss their performance in HMFOR electrocatalysis. According to the Reviewer's comment, we have examined their electrochemical HMFOR activities, including SC-MHEO-(CoNiMnCuFe)₃O₄, SC-MHEO-(CoNiMnCuZnBi)₃O₄, SC-MHEO-(CoNiMnCuZnFeBi)₃O₄ and SC-MHEO-(CoNiMnCuZn)O. Corresponding results have included in Figure S27. We have added the corresponding discussion in the revised manuscript, as read: "At the same time, we have also conducted electrocatalytic HMFOR tests on other SC-MHEO nanoplates at the optimal voltage of 1.435 V (vs. RHE). Remarkably, all Co₃O₄-like electrocatalysts exhibited considerable conversion rates and selectivities due to the unique 'cocktail' effect similar to the performance on SC-MHEO-(CoNiMnCuZn)₃O₄ (Figure S27). In comparison, SC-MHEO-

(CoNiMnCuZn)O disclosed the decreased activity in HMFOR electrocatalysis, which can be attributed to the absence of valence-changing metal ions in the crystal structure.”

Figure S27. Conversion of HMF and selectivity of FDCA for HMFOR electrocatalysis by SC-MHEO (a: SC-MHEO-(CoMnNiCuFe)₃O₄; b: SC-MHEO-(CoMnNiCuZnBi)₃O₄; c: SC-MHEO-(CoMnNiCuZnFeBi)₃O₄; d: SC-MHEO-(CoNiMnCuZn)O).

Comment 3: I doubt the so-called 2D nanoplate structure because they look like 1D rod-like structure (spindle or rice). Authors should provide topologic TEM images or AFM result for their claim.

Response: We thank the Reviewer for this valuable comment. As presented in Figure 2a (low-magnification SEM image), our samples are morphologically nanoplate-like (not rod-like) with average length, center width, and thickness of 2.9 μm, 460 nm, and 75 nm. Meanwhile, TEM image seen from side view also showed a thickness of 75 nm, which was remarkably smaller than the width of 460 nm, indicating their plate-like morphology. According to the Reviewer’s comment, we have characterized the thickness of SC-MHEO by AFM, and the results showed that the thickness was approximately 78 nm. We have added the corresponding discussion in the revised manuscript, as read: “Meanwhile, atomic force microscope (AFM) image of SC-MHEO showed a typical two-dimensional plate-like morphology with an average thickness of approximately 78 nm (Figure S12).”

Figure S12. AFM image of SC-MHEO and height profile along the red line in AFM.

Comment 4: Formation mechanism of mesopores is not clear. I suggest authors should also investigate the nitrogen sorption of high-entropy basic carbonate before calcination rather than simply quote previous references.

Response: We thank the Reviewer for this comment. In this manuscript, a new mesopore-forming mechanism of the release of OH⁻ and CO₂ was presented. Typically, the formation of SC-MHEO nanoplates is the result of a careful control over the high-temperature treatment of two-dimensional SC-HE-BCSs by a BCS-to-oxide transition route. There are abundant OH⁻ and CO₃²⁻ of BCS-(CoNiMnCuZn)₂(OH)₂CO₃. During the high-temperature treatment, both H₂O and CO₂ are released accordingly, which thus self-template the formation of abundant penetrated mesopores.

According to the Reviewer's comment, we have performed N₂ sorption characterization of parent high-entropy BSCs before the calcination. Corresponding results have been included in Figure S15. Obviously, N₂ adsorption/desorption isotherms confirmed that high-entropy BSCs were structurally nonporous. We have added the corresponding discussion in the revised manuscript, as read: “Non-porous structure of BCS-(CoNiMnCuZn)₂(OH)₂CO₃ with a low BET surface area of 9.7 m² g⁻¹ is also confirmed by N₂ sorption isotherms (Figure S15).”

Figure S15. N₂ sorption isotherms of BCS-(CoNiMnCuZn)₂(OH)₂CO₃.

Comment 5: The voltage conditions for impedance testing are not clearly labeled.

Response: We thank the Reviewer for pointing out this mistake. The voltages for the impedance tests are noted in the revised manuscript, as read: “EIS plots of SC-MHEO, P-HEO, and SC-M-CO₃O₄ at -1.485 V vs RHE.”

Comment 6: The double-layer capacitance fitting of the SC-MHEO catalyst shows some deviations from the fitting line, with some points deviating from the line. It may be necessary to adjust the scanning speed to make it align completely on a straight line.

Response: We thank the Reviewer for this valuable comment. The deviations would be originated from the voltage range selection in the non-Faradaic region, causing some points to deviate from the line. To avoid this misunderstanding, we have retested the double-layer capacitance fitting curves of the SC-

MHEO catalyst. The current curves are smoother. The results have been added to Figure S23 in the revised manuscript.

***Comment 7:** The active sites in high entropy alloy oxides are mentioned multiple times in this article, but the investigation of these active sites is not clear. It is uncertain whether the active sites are located on certain elements or on the bridging sites between two elements. Additionally, whether it is possible to calculate the Gibbs free energy of each element in the catalytic process by theoretical calculation?*

Response: We kindly thank the Reviewer for the important comments. The first results on multicomponent high-entropy materials (HEMs) were published in 2004 (Adv. Eng. Mater. 2004, 6, 74; Mater. Sci. Eng. A 2004, 375-377, 213). Generally, the HEMs were defined as "the substances consisting of five or more principal elements in equimolar ratios at concentrations between 35 and 5 at. % of each element". The multimetallization of HEMs has provided several important "core effects", including high entropy effect, lattice distortion effect, and "cocktail" effect (Energy Environ. Sci. 2021, 14, 2883). These synergistic effects were impossible to be achieved by low-entropy materials (Angew. Chem. Int. Ed. 2022, 61, e202200889; J. Am. Chem. Soc. 2022, 144, 15944; Adv. Mater. 2020, 32, 2000385; Nat. Commun. 2022, 13, 5065; J. Am. Chem. Soc. 2023, 145, 20, 11140). As mentioned by Prof. Miracle: "Unlike the other 'core effects', the 'cocktail' effect of HEMs is not a hypothesis and requires no proof. The 'cocktail effect' reminds us that exceptional materials properties often result from unexpected synergies." (Acta Mater. 2017, 122, 448). Therefore, the overall effect of high-entropy materials is unpredictable, and it is impossible to specifically determine where the active sites are located on certain components. Exploring the overall catalyst activity through reasonable establishment of models is currently the most commonly method.

To respond the Reviewer's comment, we have also used the DFT calculation to explore the active sites for HMFOR electrocatalysis in this work. First, according to existing reports, the adsorption energy of the substrate molecule (HMF) is the activity descriptor of electrochemical HMFOR (Adv. Mater. 2024, 36, 2311698; Chem. Eng. J. 2022, 433, 133842). Based on the fact, we have calculated the adsorption energies of HMF substrate molecules on different metal sites of SC-MHEO, and confirmed that the Co site has the highest adsorption energy for HMF. Corresponding results have been included in Figure S34. Therefore, Co sites are considered as the main active sites for HMFOR electrocatalysis in this work. In addition, we have also calculated the *d* band center of each metal in SC-MHEO. The results showed that the *d* band center of Co (-2.45 eV) is closer to the Fermi level than other metals (Ni:4.92 eV, Cu: 3.11 eV, Mn: 5.07 eV, Zn: 6.44 eV). This further suggested that Co is the best adsorption site for HMF. Corresponding results have also been included in Figure S35. Subsequently, based on the rational exploration of this model, we have calculated the reaction pathway diagram of SC-MHEO in the catalytic process with the Co site as the active center. The results have revealed that SC-MHEO simultaneously achieved high activity and selectivity in the electrocatalytic HMFOR process thanks to the 'cocktail' effect. The *d* band center of Co in the SC-MHEO structure is closer to the Fermi level than that of SC-M-Co₃O₄, indicating that the adsorption of HMF is stronger than that of SC-M-Co₃O₄. This is consistent with the trend of the reaction path diagram. Finally, under the influence of other elements, the peak pattern of Co PDOS in the SC-

MHEO structure tended to be more numerous and broadened compared to the SC-M-Co₃O₄. This is a typical impact of a high-entropy system on the central electronic structure.

On the basis of above discussion, it has clearly revealed that there is a typical cocktail effect in the structure of our SC-MHEO electrocatalyst. To make our statements clearer, we have added the corresponding discussion in the revised manuscript, as read: “The adsorption energy of substrate molecule (HMF) was the activity descriptor of electrochemical HMFOR. We thus calculated the adsorption energies of HMF substrate molecule on different metal sites of SC-MHEO. As presented in **Figure S34**, the Co site has the strongest adsorption of HMF (-1.77 eV), which thus was considered as the main active site for HMFOR electrocatalysis. In addition, we also calculated the *d*-band center of metal elements in SC-MHEO. The results showed that the *d*-band center of Co (-2.45 eV) was larger than that of other metals (Ni: 4.92 eV, Cu: 3.11 eV, Mn: 5.07 eV, Zn: 6.44 eV) closer to the Fermi level (**Figure S35**). It showed that the Co site had a strong ability to capture reaction intermediates, which further supported our conclusion.”

Figure S34. Adsorption model structure of HMF at (a) Co, (b) Cu, (c) Ni, (d) Mn, and (e) Zn sites of SC-MHEO. (f) Calculated adsorption energies of Co, Cu, Ni, Mn, and Zn sites in the SC-MHEO structure.

Figure S35. PDOSs of SC-MHEO.

In addition, the *d* band center of Co in SC-MHEO is closer to the Fermi level than that of SC-M-Co₃O₄, indicating a strong ability to capture reaction intermediates, which is consistent with our reaction pathway diagram. Under the influence of other elements, the peak patterns of Co partial projected density of states (PDOS) tend to be more numerous and broader in the SC-MHEO structure than SC-M-Co₃O₄ structure. This illustrates the impact of high-entropy systems on the central electronic structure (**Figure S37**).

Figure S37. Co PDOS of SC-MHEO and SC-M-Co₃O₄.

Comment 8: Can a Flow Cell be assembled for the HMFOR reaction to explore its commercial prospects?

Response: We thank the Reviewer for this valuable comment. We have added the corresponding experiments discussion in the revised manuscript, as read: “In addition, a two-electrode coupling system continuous flow electrolyzer was used to evaluate the practicality of SC-MHEO cathode for HMFOR electrocatalysis (**Figure S44**). Impressively, SC-MHEO electrocatalyst discloses a superior selectivity of 98.3%, a high FDCA yield of 87.5%, and a remarkable FE of 86.1% in a continuous flow electrolyzer. The result further highlights the potential application of SC-MHEO in real flow electrolyzer for producing high value-added chemicals.”

Figure S44. (a) LSV curve of SC-MHEO electrocatalyst in the two-electrode coupling system continuous flow electrolyzer. Inset in (a) is the two-electrode coupling system continuous flow electrolyzer. (b) Conversion, selectivity and FE of SC-MHEO cathode in the two-electrode coupling system continuous flow electrolyzer at cell voltage of 2.1 V.

Comment 9: *Is there a formula to calculate the configuration entropy? Does it comply with the standard entropy value of high entropy alloys?*

Response: We thank the Reviewer for this comment. As commented by the Reviewer, there is a formula to calculate the configuration entropy of our MHEO materials. According to the formula, we have calculated the configurational entropy of SC-MHEO. We have also added the corresponding discussion in the revised manuscript, as read: “After the calculation, the configurational entropies (S_{config}) of SC-MHEO are $>1.5R$, further indicating they are high-entropy materials (Table S3).”

Meanwhile, we have included the details of how to calculate the configuration entropy in the revised Supporting Information, as read: “**Calculation of S_{config} .**

$$S_{\text{config}} = -R \left[\left(\sum_{a=1}^n x_a \ln x_a \right)_{\text{cation-site}} + \left(\sum_{b=1}^m x_b \ln x_b \right)_{\text{anion-site}} \right]$$

n and m are the number of element types, x_a and x_b are the mole fractions of element components at cation sites and anion sites, and R is the universal gas constant ($R=8.314 \text{ J K}^{-1} \text{ mol}^{-1}$). The molar fraction used for entropy calculation was obtained from the ICP results. Materials with $S_{\text{config}} \geq 1.5 R$ are considered high-entropy systems, while materials with $1.5 R > S_{\text{config}} \geq 1 R$ and $S_{\text{config}} < 1 R$ are considered medium-entropy and low-entropy systems, respectively.”

Table S3. Calculated S_{config} values of SC-MHEO nanoplates.

Cat.	S_{config}
SC-MHEO- (CoMnNiCuZn) ₃ O ₄	1.5R
SC-MHEO- (CoMnNiCuFe) ₃ O ₄	1.6R
SC-MHEO- (CoMnNiCuZnBi) ₃ O ₄	1.6R
SC-MHEO- (CoMnNiCuZnFeBi) ₃ O ₄	1.8R
SC-MHEO- (CoMnCuNiZn)O	1.5R

Comment 10: *The superiority of mesoporous materials has been emphasized throughout this paper, and the existence of a large number of defects and oxygen vacancies in mesopores has been mentioned. However, the mechanism of how the pores affect its performance has not been thoroughly explored, and it is unclear whether the composition of elements inside the pores is consistent with that of the shell.*

Response: We thank the Reviewer for this comment. In this manuscript, the core effect brought by the high-entropy component is the main reason for the improved performance. Therefore, we focused the exploration of catalytic mechanism on the effects brought by high-entropy components. By contrast,

mesoporous structures can kinetically facilitate the transport of related species (such as protons, electrons, water, etc.) and thus boost catalytic kinetic process. In addition, mesoporous structure can also speed up the reaction rate by exposing more active sites. Finally, mesoporous structure can also increase the number of oxygen vacancies. These structural advantages can assist the catalytic process.

For compositional distribution, we think that all the elements homogeneously distribute in the nanoplates, including inside the pores and the shells. On the one hand, the nanoplates are single-crystalline, which confirms a similar crystalline in all the nanoplates. On the other hand, both XRD and STEM EDX mapping images show single set of signals, further highlighting their uniform element distributions. Uniform elemental distributions are originated from our BSC-to-oxide transition method.

***Comment 11:** SC-MHEO exhibits the highest conversion rate and Faradaic efficiency at a voltage of 1.435V, but in terms of performance at 1.435V, its advantage over P-HEO is too small.*

Response: We thank the Reviewer for this comment. SC-MHEO exhibits a conversion of 99.3% and an FDCA Faradaic efficiency of 97.7% at 1.435V. In comparison, the conversion of P-HEO catalyst is only 87.2% at 1.435V, and the FE of FDCA is only 84.9%. Excluding the influence of component elements, nearly 100% of conversion and 97.7% of selectivity can be achieved through structural control. We think that this is a significant improvement in catalytic performance.

***Comment 12:** The FT pattern in Fig S18J shows different crystal orientation. Could authors give an explanation?*

Response: We thank the Reviewer for pointing out this mistake. The FT image in Figure S18b is a mistake due to our carelessness. We have corrected the image accordingly.

Reviewer 3

Overall Comments: This work demonstrated a strategy to synthesis two-dimensional mesoporous high-entropy transition-metal oxides by thermally converting 2D high-entropy basic carbonate salts. Comparing with the traditional polycrystalline high-entropy oxides, this structure presented by the authors demonstrates a better HMF electrocatalytic oxidation performance. While I found the synthesis strategy interesting and the relevant TEM analysis attractive, the novelty of the mesoporous and single-crystal HEO as demonstrated by the author cannot satisfy the criterion of Nature Commun. Below are listed several comments to further improve this work:

Response: We kindly thank the Reviewer for reading through and providing some positive comments on our manuscript, especially in synthesis strategy and characterization of our new materials. As responded to Reviewer 2, **we combine single crystals and mesopores, which have been widely proven in electrocatalysis to significantly improve the mass and charge transfer process and catalytic stability, into high-entropy materials.** As Xie et al. commented on this type of material, "Porous single crystals at the macroscale combine the advantages of porous materials and single crystals to incorporate both porosity and structural coherence in a porous architecture, leading to invaluable opportunities to alter the material's properties by controlling the unique structural features to enhance its performance." **To the best of our knowledge, low-dimensional single-crystalline mesoporous HEOs with well-defined nanostructures and controlled metal compositions have never been achieved thus far.** Besides, our BSC template transition strategy can be easily extended to prepare other high-entropy materials with controlled compositional functions, including alloys, sulfides, phosphides, nitrides. Therefore, we strongly believe that single crystalline mesoporous HEOs and other materials are structurally novel for materials discovery and property optimization.

In terms of catalysis, our SC-MHEO-(CoNiMnCuZn)₃O₄ is demonstrated as an electrocatalyst for HMFOR, which discloses excellent HMF conversion and FDCA selectivity. All detailed pathways and reaction kinetics are carefully discussed in the revised manuscript. Meanwhile, flow cell experiments further demonstrates its potential commercial value. More impressively, we innovatively propose that SC-MHEO as bifunctional electrocatalyst that couples anode HMFOR and cathode nitrate reduction as to realize concurrent electrosynthesis of value-added FDCA and ammonia with a low energy consumption. *Considering abundant anode/cathode reactions in water, some new applications that couple different oxidation and reduction reactions into value-added products are highly desired.* We believe that the revised manuscript has achieved the breakthroughs in terms of innovation in material syntheses, property optimization, and application exploration. Therefore, we expect that the Reviewer can agree with our statements and recommend the acceptance of our work for the publication in *Nat. Commun.*

Comment 1: *The author presented several potentially attractive properties of their HEO structure in the introduction, e.g., 2D, single-crystalline, mesoporous ... Which characteristic mostly benefits the electrochemical performance and why? The reviewer cannot quickly get the designing logic of the HEO structure.*

Response: We thank the Reviewer for this comment. An ideal metal electrocatalyst generally requires abundant active sites, high amount of undercoordinated metal sites, fast electron and mass transfer ability. Here, two-dimensional single-crystalline mesoporous HEOs provide an opportunity to combine above structural advantages in one material and thus enable high performance in electrocatalysis.

Typically, two-dimensional materials have maximal surface to bulk ratios, providing a high density of surface-active sites, which is beneficial for surface-active applications. In addition, two-dimensional materials can also effectively improve the electron mobility and exhibit the excellent conductivity during electrocatalysis (*Nat. Catal.* **2018**, 1, 909; *Chem. Soc. Rev.* **2021**, 50, 12744). Meanwhile, the introduction of mesopores is beneficial to enhance mass transfer and provide accessible undercoordinated metal sites around the mesopores (*Coord. Chem. Rev.* **2022**, 466, 214604; *Nat. Rev. Mater.* **2016**, 1, 16023). Besides, single-crystalline materials have high particle integrity and few grain boundaries, which reduce side reactions between the material and the electrolyte. In addition, single-crystalline materials inhibit rapid oxidation of metals in the oxide structure and thus improve structural stability for electrocatalysis (*Acc. Chem. Res.* **2023**, 56, 3, 374; *Nat. Commun.* **2019**, 10, 3168). As discussed above, both crystal structure and surface properties are important when designing and manufacturing compounds to achieve the desired properties in a specific application. In this manuscript, we demonstrate that two-dimensional single-crystalline mesoporous HEOs combined synergistic advantages exhibit high activity, selectivity and stability in HMFOR electrocatalysis compared to P-HEO, further highlighting the importance in engineering their structure and crystallinity.

Comment 2: 1. P5/line 130: Please explain more about the XRD comparison between monometallic Co_3O_4 and HEO. Why can the peak shift indicate the replacement of Co by other transition-metals?

Response: We thank the Reviewer for this comment. The peak shifts in XRD generally originated from the changes of d -spacing distances of the nanocrystalline. In this work, Co sites in Co_3O_4 were partially substituted by other four metals (Mn, Ni, Cu, Zn). In comparison to Co atoms (72 pm), the radii of Zn (74 pm) and Mn (80 pm) are larger, resulting in an increase of d -spacing distance from 72 pm to 75 pm. According to the Bragg's law, the diffraction peaks would shift correspondingly toward the lower degrees. The results thus confirmed that Co atoms was partially substituted by other transition metals. The shift in peak is a very general phenomenon in crystalline chemistry.

Comment 3: Line 133: the crystalline structure cannot be confirmed by Raman.

Response: We thank the Reviewer for this comment. Raman spectroscopy is a spectroscopic technique that is generally used to analyze the structures and chemical compositions of matter. It is based on the phenomenon of Raman scattering and obtains the information by measuring the frequency changes of light scattered by a sample. When a laser beam shines onto a sample, some of the photons interact with molecules in the sample, causing the energy of the photons to change, thus producing scattered light. Raman spectroscopy can provide important information about the vibration, rotation, and lattice structure of molecules in a sample by measuring the changes in the frequency of scattered light. For example, Liu et al. detected the crystal structure change from V- Ni_3S_2 to $\text{Ni}(\text{OH})_2$ and then to NiOOH through Raman spectroscopy (*Nano Lett.* 2023, 23, 11, 5027). Abedzadeh et al. confirmed that their synthesized tungsten

oxide dihydrate nanosheets were monoclinic $\text{WO}_3 \cdot 2(\text{H}_2\text{O})$ phase through Raman spectroscopy analysis (Int. J. Hydrogen Energy 2024, 53, 11, 749). Gu et al. characterized the spinel phase structure ($Fd-3m$) by Raman spectroscopy. In this work, Raman spectrum showed three characteristics: E_g (350cm^{-1}), F_{2g} (515cm^{-1}), and A_{1g} (675cm^{-1}), which were attributed to the characteristic vibration modes of the $Fd-3m$ space group. Therefore, the characteristic vibration modes and lattice structures of molecules in some samples have been confirmed by Raman spectroscopy. In this work, crystalline structure of SC-MHEO has been characterized by XRD, XPS, TEM, etc. Meanwhile, Raman spectroscopy was chosen as a supplementary route to present its crystalline structure.

Comment 4: *I post my concern on the XPS analysis: the band width of O 1s peak was not consistent in SC-MHEO and P-HEO.*

Response: We thank the Reviewer for pointing out this inconsistency. Here, we think that the inconsistency in the band width may be caused by different instrument tests. To avoid this misunderstanding, we have retested the O 1s spectrum of P-HEO and found that the retested signals exhibited the similar band width of O 1s peak. The results have also been corrected in the revised manuscript accordingly.

Comment 5: *The author should present the ECSA-normalized current density. How will the mesoporous structure contribute to the performance, only by increasing the exposure of active sites?*

Response: We thank the Reviewer for this comment. According to the Reviewer's comment, we have added the ECSA-normalized current density in the revised manuscript (Figure S24). Obviously, the ECSA normalized current density showed a smaller difference in activity between SC-MHEO and P-HEO. However, SC-MHEO still exhibited the lower current density, indicating its better activity. This is mostly because of mesoporous structure that exposes more undercoordinated metal sites and produces defected or uncoordinated oxygen atoms (generally more active) for electrocatalysis. Considering the abundant metal and oxygen sites, mesoporous structure would present more structural advantages. This is why we designed mesoporous HEOs for electrocatalysis. We have added the corresponding discussion in the revised manuscript, as read: "ECSA normalization of current density of SC-MHEO and P-HEO is also compared in 50 mM HMF, indicating mesoporous structure exposes more undercoordinated metal and oxygen sites that further promotes HMFOR electrocatalysis (Figure S24)"

Figure S24. ECSA-normalized polarization curves of SC-MHEO in 50 mM HMF.

Comment 6: *Will the single-crystalline nature contribute to the excellent performance?*

Response: We thank the Reviewer for this comment. As responded in Comment 1, single-crystalline structure has high particle integrity and few grain boundaries, which thus reduces the side reactions between the material and the electrolyte. In addition, the single-crystalline materials inhibit rapid oxidation of metals in the oxide structure thus improve structural stability. In this work, single-crystalline structure plays a key role in electron transfer and catalytic stability during the oxidation process.

Comment 7: *A thorough performance comparison with literatures should be listed in the manuscript. General: The words should be polished extensively. Many typos can be found throughout the current paper. For example, Line 213-214; Line 228-229...*

Response: We thank the Reviewer for the comments. First, we have included a thorough performance comparison with the literature in Table S4 of Supporting Information due to the limited space in Manuscript. Besides, according to the Reviewer's comment, we have double-checked and revised the manuscript thoroughly. All the mistakes, including typos and others, have been changed accordingly.

Reviewer 4

Overall Comments: This manuscript reports a very facile solid-phase synthetic strategy for preparing high-quality mesoporous high-entropy oxides (MHEOs). By the utilization of pre-synthesized high-entropy basic carbonates as the parent template, in situ conversion and release of H₂O and CO₂ result in precise synthesis of high-quality MHEOs with two-dimensional and single-crystalline structures. Meanwhile, this strategy is universal and can be extended to prepare other two-dimensional MHEOs with controlled elemental compositions and phase structures. This should be the first work on preparing two-dimensional single-crystalline MHEOs (SC-MHEOs). In the catalysis part, the authors explore the performance of SC-MHEOs in a two-electrode coupling system. SC-MHEO-(CoNiMnCuZn)₃O₄ nanoplates exhibit excellent activity and stability in both the HMFOR reaction at the anode and the NO₃⁻RR reaction at the cathode.

In overall, the manuscript is well-organized and written. The conclusions are also supported well by carefully conducted experiments. I believe the work would be of interest for the broad readership of the journal and will make a significant contribution to the field of two-dimensional high-entropy electrocatalysts and beyond. In my opinion, this manuscript can be accepted for publication in Nature Communications with the following revisions.

Response: We kindly thank the Reviewer for reading through and providing such positive comments on our manuscript.

Comment 1: *The author mentioned in the manuscript "In comparison to parent basic carbonate salt, SC-MHEO becomes slightly smaller with average length, center width, and thickness of 2.9 μm, 460 nm, and 70 nm, indicating a minor volume shrinkage during the calcination." However, low-magnification SEM cannot clearly explain this problem. Structural information similar to Figure S3 needs to be supplemented.*

Response: We thank the Reviewer for this comment. We have marked the average lengths, center widths, and thicknesses of SC-MHEO nanoplates and their parent basic carbonate salts by recording in high-magnification SEM images. Typically, the average lengths, center widths, and thicknesses of basic carbonate salts are 3.1 μm, 480 nm, and 90 nm, respectively. In sharp comparison, SC-MHEO becomes slightly smaller with average length, center width, and thickness of 2.9 μm, 460 nm, and 75 nm, respectively. According to the Reviewer's comment, we have included detailed characterizations of SC-MHEO in Figures S11 and S12 by TEM and AFM images. The results have clearly revealed a minor volume shrinkage during the BCS-to-oxide transition process.

Comment 2: *HMFOR and OER are two competing reactions. The authors should provide the theoretical basis for the preferential occurrence of the HMFOR reaction.*

Response: We thank the Reviewer for this comment. We have carefully considered all possible water oxidation side reactions in aqueous solution, including 4-electron oxygen evolution reaction (4e⁻ OER, product is H₂O) and 2-electron water oxidation reaction (2e⁻ WOR, product is H₂O₂). Corresponding results have been included in Figure S39. We have also added the corresponding discussion in the revised

manuscript, as read: “In addition, HMFOR and oxygen evolution reaction (OER) are two competing reactions due to possible water oxidation side reactions in aqueous solution. Gibbs free energies of two pathways, including the 4-electron oxygen evolution reaction and the 2-electron water oxidation reaction, are calculated accordingly (**Figure S38**). The results show that the reaction energy of the first step of both reactions ($\text{H}_2\text{O} \rightarrow \text{OH}^* + \text{H}^+ + \text{e}^-$) reaches +1.54 eV, which is much higher than any step in the reaction pathway in HMFOR electrocatalysis. Therefore, the two side reactions do not occur preferentially.”

Figure S38. Gibbs free energy diagrams and reaction paths for OER and WOR electrocatalysis of SC-MHEO.

Comment 3: *The author needs to identify the source of N in NH_3 in the NO_3^- RR reaction at the cathode, such as ^{15}N isotope labeling experiments*

Response: We thank the Reviewer for this valuable comment. According to the Reviewer’s comment, ^{15}N isotope labeling experiments have been included to the revised manuscript. We have added the corresponding content in the revised manuscript, as read: “The origin of produced NH_3 was identified through the ^{15}N isotope labeling experiments. The typical $^{15}\text{NH}_3$ peak can be seen when using $^{15}\text{NO}_3^-$ as nitrogen source, indicating that the NH_3 produced comes from NO_3^- RR (**Figure S40**).”

Figure S40. ^1H NMR spectra of the electrolyte after electrocatalytic NO_3^- RR using $^{15}\text{NO}_3^-$ and $^{14}\text{NO}_3^-$ as the nitrogen source.

Comment 4: *In this manuscript, the pores were produced by solid-phase high-temperature treatment for the release of H_2O and CO_2 . Did the authors tune the treatment temperatures? Can pore properties (sizes) be changed by temperatures?*

Response: We thank the Reviewer for this valuable comment. As the Reviewer mentioned, the formation of mesopores relies only on the removal of small molecules (such as CO_2 and H_2O) produced during the thermal decomposition of carbonate hydroxides. In fact, we have changed the temperatures to regulate the pore size of SC-MHEO. When the calcination temperature reaches 260 $^\circ\text{C}$, mesoporous structure is initially formed. As the temperature increases, the crystallinity increases and the pore size gradually becomes larger. As the temperature increased to 400 $^\circ\text{C}$, the cracks begin to appear in the pore. When the calcination temperature reaches 550 $^\circ\text{C}$, mesoporous structure is completely destroyed. Based on the experiments, we selected the calcination temperature of 300 $^\circ\text{C}$ and the calcination time of 3 h, which not only maintained the integrity of mesoporous structure but also ensured the strong crystallinity of the materials.

Comment 5: *During the NO_3^- RR stability test, the yield of NH_3 increased after 12 cycles of testing. What is the reason?*

Response: We thank the Reviewer for this comment. Due to the high structural functions, our SC-MHEO electrocatalyst is highly stable for NO_3^- RR. Meanwhile, NH_3 yield rate is pretty high, so we think the change in activity is not obvious (It generally happens in NO_3^- RR electrocatalysis). To make our statement clearer, we have also performed structural characterizations of SC-MHEO after stability test. All the results have included in Figure S42. Obviously, SC-MHEO retains well the structure and crystallinity after stability test. We have added the corresponding content in the revised manuscript, as read: “Physical characterizations also show that the structure and crystallinity of SC-MHEO catalyst retains well (Figure S41).”

Figure S41. (a) HRTEM image, (b) TEM image and (c) corresponding FT patterns of SC-MHEO after the stability test.

Comment 6: *In Figure 6b, the authors mentioned that NH_3 yield of SC-MHEO showed a typical volcanic trend in the potential range from -0.30 V to -0.55 V. However, this rule is not followed at -0.50 V and -0.55 V. The author should test the NH_3 yield at more negative voltages.*

Response: We thank the Reviewer for this comment. As presented by the Reviewer, there is a small change in the volcanic trend of FE_{NH_3} (-0.50 V and -0.55 V). This is mostly because of the competitive relationship between NO_3^-RR and HER under the high voltage. Generally, in the higher voltage, HER is weaker, resulting in a slightly higher FE_{NH_3} accordingly. This is consistent with many reported results (For example: Angew. Chem. Int. Ed.2023,62, e2022187; Adv. Energy Mater.2023,13, 2302274; Angew. Chem. Int. Ed. 2023, 62, e202303327).

REVIEWERS' COMMENTS

Reviewer #1 (Remarks to the Author):

I think the authors have addressed my comments appropriately. The quality of this work has been improved. I recommend acceptance as it is.

Reviewer #2 (Remarks to the Author):

This revised manuscript has addressed my concerns.

Reviewer #3 (Remarks to the Author):

The author has addressed my previous comments, and I think it is suitable to be published on Nat. Commun.

Reviewer #4 (Remarks to the Author):

The authors have addressed all my comments and I think it can be accepted without revision.